# PHYSICS-GUIDED LEARNING OF METEOROLOGICAL DYNAMICS FOR WEATHER FORECASTING AND DOWNSCALING

## ABSTRACT

Weather forecasting is of paramount importance for a myriad of societal and scientific applications. Traditionally, numerical weather prediction (NWP) methods based on physical principles are computationally intensive and can struggle with the inherent complexity of atmospheric dynamics. Recently, deep learning techniques have shown promise in weather prediction, but the long-term generalization and physical consistency of pure data-driven approaches remain challenging. In this paper, we introduce a novel physics-guided approach for numerical weather prediction that combines the strengths of both physical mechanism and deep learning, namely *PhyDL-NWP*. Our method can capture the nonlinear dynamics of meteorology and align deep learning models with the underlying physical mechanism to improve generalization. Extensive experiments on real-world weather datasets show that our model can significantly improve the performance of deep learning methods in a wide range of tasks from forecasting to downscaling.

## 1 INTRODUCTION

Weather forecasting remains one of modern science's most complex and vital challenges, with far-reaching implications for agriculture, renewable energy, disaster management, transportation, etc. The first-principle approach to weather forecasting, i.e., numerical weather prediction (NWP), relies on mathematical models of atmospheric and oceanic phenomena. NWP is often computationally intensive and can struggle with the inherent complexity of atmospheric dynamics, which arises from the nonlinear interactions between various meteorological variables, the vast spatial and temporal scales involved, and the chaotic nature of weather systems (Holton, 1973). For example, while the fundamental equations like the Navier-Stokes equations are well understood, it remains challenging to model turbulence accurately due to certain assumptions made to simplify the system and make them solvable within reasonable time (Stull, 1988). The inevitable simplifications and approximations can introduce inaccuracies, some of which involve linearizing nonlinear or parametric terms, making assumptions about the symmetry of flow, or ignoring small-scale interactions. For instance, the hydrostatic approximation (Phillips, 1966) used in NWP assumes a balance between vertical pressure forces and gravity, ignoring vertical acceleration, which can break down for small-scale phenomena (White & Bromley, 1995). The Boussinesq approximation (Spiegel & Veronis, 1960) assumes negligible density variations except in buoyancy terms, which can be an oversimplification in atmospheric modeling where density variations might play a significant role.

In the context of machine learning and particularly deep learning, there has been a growing interest in weather prediction as well. Deep learning models, with the ability to capture complex nonlinear relationships, have shown promises in tasks such as weather forecasting (Pathak et al., 2022; Bi et al., 2023; Wu et al., 2023) and downscaling (Vandal et al., 2017). However, the application of deep learning to weather prediction is not without challenges. One of the key issues is the generalization ability of pure data-driven approaches. While deep learning models can excel at fitting complex patterns in training data, they lack the ability to generalize well to unseen scenarios by capturing noise or specificities of the training data. Moreover, these models often do not consider the physical mechanisms that govern weather systems, leading to predictions that may be statistically accurate for the training dataset but physically inconsistent.

Numerical Weather Prediction (NWP) and deep learning represent two distinct paradigms in weather forecasting, each with its unique strengths and challenges. By combining the strengths of both sides, a physics-guided deep learning model could offer a more holistic solution. In the literature, Physics-Informed Neural Networks (PINN) (Raissi et al., 2019) have been widely used as an alternative for numerical simulation. However, weather prediction is highly complicated, with many weather factors and physical processes in the play, easily influenced by local variations like change of boundary conditions, small-scale phenomena like microclimates and external forces like heat from the sun. Many of these vital factors, which have huge impacts in the first-principle equations, are missing in the data due to difficulties to measure and quantify. Without a complete equation, PINN will be hard to train. In light of this problem, we propose a novel framework called PhyDL-NWP. Inspired by (Rudy et al., 2017), We create a library of partial differential terms relevant to the existing physical equations for related weather factors, which allows us to obtain a complete equation that reflects the physical mechanism of meteorology in the data, while aligning the deep learning model with the physical mechanism. Moreover, we propose a latent force model as a parametrization term to supplement the forces that cannot be represented by the selected explicit PDE terms, following the parametrization strategy (Warner, 2010) widely adopted in the literature of meteorology.

This paradigm first trains a neural network to predict weather conditions given the spatio-temporal coordinates and then uses the auto-differentiation of this network to obtain the partial differential equation (PDE) terms. Using this paradigm of PhyDL-NWP, we prove to achieve state-of-the-art performances in weather downscaling, with additional advantages of unlimited granularity and physical understanding. Based on that, with PDEs to reflect the physical mechanism as an amendment to the physics theory for understanding the data, we are able to constrain the optimization of deep learning models and improve its generalization across different environments. In extensive experiments on 13 baselines and three datasets, we find that this knowledge discovery process is effective and many resultant physics-guided deep learning models are more accurate than their vanilla models.

In short, our contributions are summarized as follows:

- We propose a physics-guided learning framework *PhyDL-NWP* that incorporates physical knowledge to improve our understanding of the physical mechanism of meteorology and the generalization ability of deep learning models for weather prediction.

- The proposed method can be used as a plug-and-play module to align deep learning models with physical consistency for a variety of tasks, ranging from medium-range weather forecasting with continuous spacetime to weather downscaling with unlimited granularity. PhyDL-NWP is very efficient to train and, with only up to 60 thousand parameters.

- Extensive experiments on real-world datasets and baselines show that our method can provide significant performance improvement for state-of-the-art models and provide insights to understand the underlying physical mechanism.

## 2 RELATED WORKS

### 2.1 WEATHER PREDICTION

In the area of weather prediction (Alley et al., 2019), Numerical Weather Prediction (NWP) (Lorenc, 1986; Bauer et al., 2015) is the current mainstream method. It uses mathematical models of the atmosphere and oceans, such as partial differential equations (PDE), to predict future weather based on current weather conditions. Some notable NWP models include European Centre for Medium-Range Weather Forecasts (ECMWF) [1], Global Forecast System (GFS) [2], etc. NWP can forecast weather in the medium range (beyond a few days ahead) but usually involves extensive computation. For example, ECMWF operates one of the largest supercomputer complexes in Europe [3].

In the recent few years, deep learning has emerged as another promising solution to weather forecasting (Hu et al., 2021) and downscaling (Vandal et al., 2017) tasks, owing to its incredible ability to model complex nonlinear relationships. These deep learning models (Wang et al., 2019; Han et al.,

---

[1] https://www.ecmwf.int/

[2] https://www.ncei.noaa.gov/products/weather-climate-models/global-forecast

[3] https://www.ecmwf.int/en/about/who-we-are

2022) mainly rely on different neural architectures such as LSTM (Li et al., 2022), CNN (Weyn et al., 2020), GNN (Lin et al., 2022; Keisler, 2022) and Transformer (Wu et al., 2023) to capture the evolving dynamics and correlation across space and time. Many large models emerge in recent years. For example, ClimaX (Nguyen et al., 2023a), GraphCastNet (Lam et al., 2022), ClimateLearn (Nguyen et al., 2023b), FengWu (Chen et al., 2023), Pangu-Weather (Bi et al., 2023) all use backbones such as the Vision Transformer (ViT) (Dosovitskiy et al., 2020), UNet (Ronneberger et al., 2015) and autoencoders, for training a large model for weather forecasting. WeatherBench (Rasp et al., 2020; 2023) benchmarks the use of pre-training techniques for weather forecasting. In addition, Fourcast-Net (Pathak et al., 2022) leverages the adaptive Fourier neural operator (AFNO) (Li et al., 2020; Guibas et al., 2021) to treat weather as a latent PDE system. However, these studies rarely consider the underlying physical mechanisms in weather prediction that are globally consistent across environments.

## 2.2 SPATIO-TEMPORAL MODELING

Spatio-temporal modeling based on deep learning (Yu et al., 2021a) has thrived in recent years, showing promising results in various applications such as video recognition (Cai et al., 2021), traffic flow forecasting (Luo et al., 2019), disease spread modeling (Arenas et al., 2020), etc. Weather prediction shares striking similarities with these applications. Thus, many previous works (Moosavi et al., 2019; Castro et al., 2021; Han et al., 2021) also approach the weather prediction task from the perspective of spatio-temporal modeling. Many models such as recurrent neural networks (Liu et al., 2016; Wang et al., 2022), convolutional neural networks (Tran et al., 2015; Xu et al., 2019), graph networks (Yu et al., 2018; Geng et al., 2019), transformers (Law & Lucas, 2022) and hybrid models (Yan et al., 2021) are extensively applied to capture the correlations of variables across space and time.

## 2.3 DYNAMICAL SYSTEM MODELING

Dynamical systems modeling (Morton et al., 2018; Long et al., 2018; Li et al., 2019) involves the mathematical formulation of systems whose states evolve over time, which often takes the form of differential equations for continuous systems. Given the governing equations, physics-informed approaches (Raissi et al., 2019; Karniadakis et al., 2021) use the physics mechanism to enhance the dynamical systems. In the absence of governing equations, the discovery of physical equations (Brunton et al., 2016; Seo et al., 2019) are proposed to explain the observation data and provide theoretical insights with respect to laws of physics (Wu & Tegmark, 2019; Iten et al., 2020). The scientific discovery of partial differential equations (PDEs) based on symbolic learning (Schmidt & Lipson, 2009; Chen et al., 2021b) and sparse regression (Rudy et al., 2017; Rao et al., 2021a) has been extensively studied in recent years. Some works also incorporate the physics-informed concept to guide the knowledge discovery (Rao et al., 2021b; Chen et al., 2021a). In addition, studies on neural differential equations (Chen et al., 2018) and operators (Lu et al., 2021; Kovachki et al., 2021) that use neural networks to approximate the function of differential equations and operators also shed light on dynamical system modeling as a latent mathematical model.

## 3 METHODOLOGY

## 3.1 PRELIMINARY

The weather dataset $\mathbf{u} = [u_1(x, y, t), ..., u_h(x, y, t)]$ can be viewed as $h$ weather factor fields (e.g., temperature, pressure), each with respect to some input coordinates $(x, y, t)$, where $x \in [1, ..., n]$ and $y \in [1, ..., m]$ are the spatial coordinates and $t \in [1, ..., T]$ is the temporal coordinate. Alternatively, it can also be represented as a sequence $\mathbb{X} = [X_1, ..., X_T]$, where $X_i \in \mathbb{R}^{n \times m \times h}$. Based on the weather data, we can consider two tasks: 1) The objective of weather forecasting is to predict weather factors in the future r hours $[\mathbb{X}]_{i+1}^{i+r} = [X_{i+1}, ..., X_{i+r}]$ based on the past s+1 hours $[\mathbb{X}]_{i-s}^{i} = [X_{i-s}, ..., X_i]$ for every time $i$; 2) The objective of weather downscaling is to predict finer-granular data $\mathbb{Y} = [Y_1, ..., Y_T]$, where $Y_i \in \mathbb{R}^{n' \times m' \times h}$, based on the original coarse-granular $\mathbb{X}$, where $n' > n$ and $m' > m$.

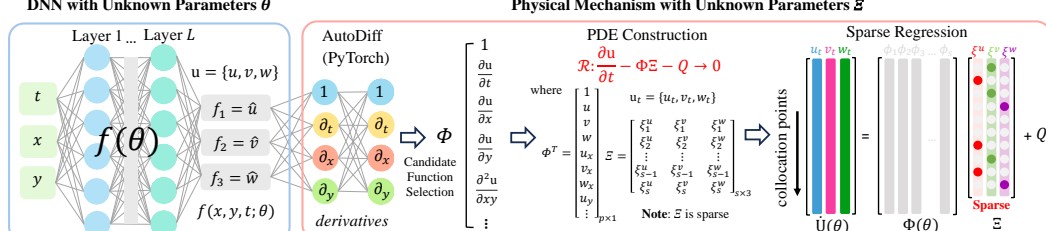

Figure 1: Schematic diagram of PhyDL-NWP for downscaling. First, given a continuous input coordinate $(x, y, t)$, the surrogate model $f_\theta$ approximates the weather data. Then, based on PyTorch's auto-differentiation and the existing meteorology theory, we calculate the derivatives for the construction of physical mechanisms driven by PDE. Last, based on sparse regression on the collocation points, we discover the PDE that fits the weather data well to provide physical guidance.

## 3.2 METEOROLOGY DYNAMICS AND WEATHER DOWNSCALING

In this paper, we aim to discover the physical mechanism that suits the weather data and correct the existing NWP. NWP is based on the numerical simulation of governing equations of meteorology. A typical partial differential equation (PDE) with parametrization has the following form:

$$\frac{\partial \mathbf{u}}{\partial t} = Q_\pi(x, y, t) + \Phi(\mathbf{u})\Xi = Q_\pi(x, y, t) + \sum_{i=1}^{p} \phi(\mathbf{u})_i \xi_i, \tag{1}$$

where

$$\phi(\mathbf{u})_i \in [1, \mathbf{u}, \frac{\partial \mathbf{u}}{\partial x}, \frac{\partial \mathbf{u}}{\partial y}, \frac{\partial^2 \mathbf{u}}{\partial x^2}, ..., \mathbf{u}\frac{\partial \mathbf{u}}{\partial x}, ...], \tag{2}$$

and $p$ denotes the number of partial derivative candidate terms considered in the task, $\phi(\mathbf{u})_i$ represents the various equation terms in the meteorology literature, with the set of $\xi_i$ as the coefficients. $Q_\pi(x, y, t)$ denotes the latent force modeled by a neural network that cannot be represented by $\Phi(\mathbf{u})$, as a supplement to missing variables and parameterization terms in the equation, such as friction.

Here, we develop a multitask deep learning model $f_\theta(x, y, t)$ as the surrogate model to the $h$ weather factor fields, which takes the spatio-temporal coordinates as input and outputs the weather factor values. The schematic diagram of the proposed surrogate model is shown in Fig. 1.

Suppose that the deep learning model is accurate enough, we will be able to approximate each $\phi(\mathbf{u})_i$ accurately based on the auto-differentiation powered by deep learning frameworks such as PyTorch and eventually obtain the corrected PDE to explain the weather factors. Simultaneously, the updated physical mechanism provides insights to guide the optimization of the deep learning model as well, which ensures that the learned physics is consistent with the prediction of deep learning. Therefore, the overall loss function can be written as the combination of data loss and physics loss:

$$\mathcal{L}_{\text{Downscale}}(\theta, \Xi, \pi) = \mathcal{L}_{\text{data}}(\theta) + \alpha\mathcal{L}_{\text{physics}}(\theta, \Xi, \pi) + \mathcal{L}_{\text{reg}}(\theta, \Xi, \pi), \tag{3}$$

where

$$\mathcal{L}_{\text{data}}(\theta) = \frac{1}{\text{nmT}} \sum_{x,y,t} \|f_\theta - \mathbf{u}\|_2^2, \tag{4}$$

$$\mathcal{L}_{\text{physics}}(\theta, \Xi, \pi) = \frac{1}{\text{n'm'T'}} \sum_{x',y',t'} \left\| \frac{\partial f}{\partial t} - \Phi(f)\Xi - Q_\pi(x', y', t') \right\|_2^2, \tag{5}$$

$$\mathcal{L}_{\text{reg}}(\theta, \Xi, \pi) = \sigma_1 \|\theta\|_2 + \sigma_2 \|\Xi\|_0 + \sigma_3 \|Q_\pi\|_2. \tag{6}$$

Here, the data loss measures whether $f_\theta$ approximates $u$ well on the weather data, and the physical loss measures whether the discovered equation fits the weather data. The regularization loss consists of three components: 1) L2 loss to prevent the overfitting of $f_\theta$; 2) L0 loss for promoting the sparsity of PDE terms to ensure equation conciseness; and 3) L2 loss for balancing the latent force so that the differential terms $\Phi(\mathbf{u})$ can maintain sufficient expression ability. To calculate the physical loss,

Figure 2: Schematic diagram of PhyDL-NWP for forecasting. Given the historical data, we use a state-of-the-art (SOTA) model to predict future data. Based on the spatio-temporal coordinates of the predicted data, we add a physics loss to recover the previously learned PDE.

inspired by previous studies (Babuška et al., 2007; Chen et al., 2021b), we combine the coordinates of weather data $x, y, t$ with randomly sampled coordinates to create more collocation points $x', y', t'$ to improve generalization, where $x' \in [1, ..., n']$, $y' \in [1, ..., m']$ and $t' \in [1, ..., T']$.

With the trained $f_\theta$, we can easily obtain the weather factors at any given input coordinates $(x, y, t)$, which can be continuous over the spacetime. This property of modeling the meteorology dynamics is naturally suitable for weather downscaling: unlike the previous methods powered by the discrete encoding and decoding of neural networks on finer-grained data as labels, the $f_\theta$ in PhyDL-NWP can perform weather downscaling with unlimited granularity without labels given a continuous coordinate, as described by Fig. 1. Therefore, as long as we obtain the dynamics of meteorology and the solver of weather factors through updating $\Xi$ and $\theta$, the downscaling task is solved automatically.

## 3.3 Weather Forecasting

On the other hand, for weather forecasting, although $f_\theta(x, y, t)$ can also take in the future data points and produce extrapolation, it only consists of dense layers that lack the advantages of state-of-the-arts models, such as the long-term memory mechanism of LSTM and the self-attention mechanism of Transformers. In particular, $f_\theta$ is only trained on historical coordinates, while anywhere outside the bounds of where the model was trained is completely unknown to $f_\theta$. To combine the advantages of both physics and deep learning, instead of using $f_\theta$, we propose to leverage the learned physical mechanism represented by $\Xi$ to improve another state-of-the-art forecasting model $g_\omega$, which takes historical spatio-temporal data and outputs future, based on the predicted data $\hat{u}(x, y, t)$ for $i + r \geq t > i$. The overall computational loss function is:

$$\mathcal{L}_{\text{Forecast}}(\omega, \theta, \Xi, \pi) = \mathcal{L}_{\text{Downscale}}(\theta, \Xi, \pi) + \beta \mathcal{L}_{\text{data}}(\omega) + \gamma \mathcal{L}_{\text{physics}}(\omega), \tag{7}$$

where

$$\mathcal{L}_{\text{data}}(\omega) = \frac{1}{r \cdot (T\text{-r-s-1})} \sum_{i=s+1}^{T-r} \left\| g_\omega([\mathbb{X}]_{i-s}^i) - [\mathbb{X}]_{i+1}^{i+r} \right\|_2^2, \tag{8}$$

$$\mathcal{L}_{\text{physics}}(\omega) = \frac{1}{r \cdot (T\text{-r-s-1})} \sum_{i=s+1}^{T-r} \left\| \frac{\partial g_\omega([\mathbb{X}]_{i-s}^i)}{\partial t} - \Phi(g_\omega([\mathbb{X}]_{i-s}^i))\Xi \right\|_2^2. \tag{9}$$

Note that $\theta$, $\Xi$ and $\pi$ are already learned during the downscaling beforehand and remain fixed when we optimize $\mathcal{L}_{\text{physics}}(\omega)$ and $\mathcal{L}_{\text{data}}(\omega)$. Therefore, the optimization of $\omega$ does not affect $\theta$, $\Xi$ and $\pi$. Here, $\alpha, \gamma, \beta, \sigma_1, \sigma_2, \sigma_3$ are all hyperparameters to balance the different loss terms. The overall framework of weather forecasting is depicted in Fig. 2. To calculate differential terms efficiently, we use the Central Finite Difference Approximations instead of training a surrogate model for every $g_\omega([\mathbb{X}]_{i-s}^i)$. If we verify that the downscaling is accurate, the discovered physics can thus provide a globally consistent constraint to help improve the generalization of $g_\omega$. In addition, as shown in Fig. 7 in Appendix A.3, the number of parameters of *PhyDL-NWP* are much less than other state-of-the-art models.

Table 1: The RMSE comparison for weather downscaling of different models for Huadong dataset. Bold fonts mark the best performances and underlines mark the second-best performances.

| Model | 100m Wind (U) | | 10m Wind (U) | | Temperature | | Surface Pressure | | Average Factor | |
|---|---|---|---|---|---|---|---|---|---|---|
| | 2x | 4x | 2x | 4x | 2x | 4x | 2x | 4x | 2x | 4x |
| Bicubic | 1.687 | 1.765 | 1.215 | 1.272 | 1.714 | 1.848 | 0.818 | 1.220 | 1.515 | 1.654 |
| EDSR | 1.145 | 1.176 | 1.020 | 1.113 | 1.217 | 1.275 | 0.460 | 0.552 | 1.068 | 1.156 |
| ResDeepD | _1.092_ | _1.111_ | 1.003 | 1.079 | _1.182_ | _1.204_ | _0.301_ | _0.317_ | _1.010_ | _1.043_ |
| RCAN | 1.169 | 1.199 | _0.808_ | _1.038_ | 1.219 | 1.259 | 0.572 | 0.609 | 1.092 | 1.144 |
| FSRCNN | 1.197 | 1.202 | 1.090 | 1.126 | 1.198 | 1.233 | 0.430 | 0.560 | 1.093 | 1.149 |
| YNet | 1.116 | 1.125 | 0.947 | 1.103 | 1.192 | 1.226 | 0.467 | 0.575 | 1.062 | 1.125 |
| DeepSD | 1.205 | 1.216 | 1.020 | 1.117 | 1.218 | 1.265 | 0.454 | 0.591 | 1.087 | 1.149 |
| PhyDL-NWP | **0.973** | **0.970** | **0.696** | **0.693** | **0.905** | **0.904** | **0.211** | **0.216** | **0.794** | **0.789** |
| Improv | 10.9% | 12.7% | 13.9% | 33.2% | 23.4% | 24.9% | 29.9% | 31.9% | 20.2% | 24.6% |

## 4 EXPERIMENTS

We conduct both the forecasting and downscaling performance comparison. All the experiments are carried out on four NVIDIA A100 PCIe 80 GB graphical cards. Only the performances on the test sets at the optimal performance on the validation sets are reported. The maximum training epochs are 50. Every result is the average of three independent training under different random seeds. Each time, we select four representative factors and the average of all factors in the tables for visualization. Average Factor is most vital for it measures the overall performance. We use two frequently used metrics (Bi et al., 2023) for evaluation: Root Mean Square Error (RMSE) and Anomaly Correlation Coefficient (ACC).

### 4.1 DOWNSCALING PERFORMANCE COMPARISON

We evaluate the effectiveness of *PhyDL-NWP* and other baseline models for weather downscaling on a real-world dataset Huadong, which is derived from the European Centre for Medium-Range Weather Forecasts (ECMWF) operational forecast (HRES) and reanalysis (ERA5) archive. It comprises a grid of $64 \times 44$ cells, with each cell having a grid size of 0.25 degrees in both latitude and longitude. More data details can be found in Appendix A.1. Since most previous studies on weather downscaling can only handle the downscaling of the two spatial dimensions, for the sake of comparison, we also only report the performance of *PhyDL-NWP* on spatial downscaling in Table. 1. We perform 2x and 4x downscaling tasks with 0.5 and 1 degrees resolutions, respectively. To facilitate this, the 0.25-degree HRES data undergoes linear interpolation to generate the requisite 0.5-degree and 1-degree input data. We compare our model against the Bicubic interpolation (Keys, 1981), FS-RCNN (Passarella et al., 2022), ResDeepD (Sharma & Mitra, 2022), EDSR (Jiang & Chen, 2022), RCAN (Yu et al., 2021b), YNet Liu et al. (2020) and DeepSD Vandal et al. (2017). For the deep learning baselines, channel-wise normalization is performed for training efficiency. Details about baselines can be found in Appendix A.2.

From Table 1, we can conclude that *PhyDL-NWP* provides a significant improvement up to 20.2% to 24.6% on average over RMSE against the baselines models. Well-recognized deep learning models like FSRCNN seem not good at weather downscaling, probably because each convolutional block only has a local receptive field for the spatial dimensions and not for the temporal dimension. The weather data has multiple variables and the spatio-temporal dependencies are not completely local, making it difficult to recover all the ground-truth information without advanced physical modeling. Furthermore, we find that the RMSE for 2x and 4x resolutions are close. Since *PhyDL-NWP* can provide infinite resolution results given continuous coordinates, we believe that it will be accurate for higher resolution downscaling, based on this evidence. Moreover, *PhyDL-NWP* can easily perform downscaling in the temporal dimension.

### 4.2 FORECASTING PERFORMANCE COMPARISON

We evaluate the effectiveness of *PhyDL-NWP* for weather forecasting on two real-world datasets collected by ECMWF[4]: Ningbo and Ningxia, which cover two different terrain types in China. On

---

[4]https://www.ecmwf.int/en/forecasts/datasets

Table 2: Model comparison for 7-day weather forecasting for Ningbo dataset.

| Model | 100m wind | | 10m wind | | Humidity | | Temperature | | Average Factor | |
|---|---|---|---|---|---|---|---|---|---|---|
| | RMSE↓ | ACC↑ | RMSE↓ | ACC↑ | RMSE↓ | ACC↑ | RMSE↓ | ACC↑ | RMSE↓ | ACC↑ |
| NWP | 0.892 | 0.606 | 0.875 | 0.581 | 0.932 | **0.699** | 0.422 | 0.910 | 0.868 | **0.587** |
| PINN | **0.622** | 0.520 | **0.605** | 0.489 | 0.835 | 0.443 | 0.657 | 0.727 | 0.652 | 0.427 |
| Bi-LSTM-T | 0.666 | 0.588 | 0.704 | 0.562 | 0.576 | 0.597 | 0.472 | 0.876 | 0.601 | 0.443 |
| Bi-LSTM-T+ | 0.635 | **0.649** | 0.664 | 0.621 | 0.550 | 0.672 | 0.442 | 0.903 | 0.571 | 0.485 |
| Improv | 4.65% | 10.4% | 5.68% | 10.5% | 4.51% | 12.6% | 6.36% | 3.08% | 5.00% | 9.48% |
| Hybrid-CBA | 0.674 | 0.568 | 0.717 | 0.550 | 0.590 | 0.595 | 0.460 | 0.865 | 0.617 | 0.431 |
| Hybrid-CBA+ | 0.641 | 0.637 | 0.680 | 0.609 | 0.572 | 0.657 | 0.411 | 0.906 | 0.586 | 0.474 |
| Improv | 4.90% | 12.1% | 5.16% | 10.7% | 3.05% | 10.4% | 10.7% | 4.74% | 5.02% | 9.98% |
| ConvLSTM | 0.701 | 0.524 | 0.732 | 0.535 | 0.572 | 0.602 | 0.489 | 0.858 | 0.636 | 0.418 |
| ConvLSTM+ | 0.658 | 0.587 | 0.699 | 0.607 | 0.550 | 0.671 | 0.454 | 0.891 | 0.596 | 0.463 |
| Improv | 6.13% | 12.0% | 4.51% | 13.5% | 3.85% | 11.5% | 7.16% | 3.85% | 5.97% | 10.8% |
| AFNO | 0.659 | 0.592 | 0.710 | 0.546 | 0.528 | 0.584 | 0.429 | 0.894 | 0.599 | 0.465 |
| AFNO+ | 0.625 | 0.648 | 0.669 | **0.630** | 0.500 | 0.695 | 0.397 | **0.929** | 0.556 | 0.530 |
| Improv | 5.16% | 9.46% | 5.78% | 15.4% | 5.30% | 19.0% | 7.46% | 3.91% | 7.18% | 14.0% |
| MTGNN | 0.685 | 0.566 | 0.720 | 0.538 | 0.521 | 0.589 | 0.434 | 0.887 | 0.597 | 0.457 |
| MTGNN+ | 0.657 | 0.629 | 0.672 | 0.613 | **0.489** | 0.679 | **0.388** | 0.918 | **0.555** | 0.514 |
| Improv | 4.09% | 11.1% | 6.67% | 13.9% | 6.14% | 15.3% | 10.6% | 3.49% | 7.04% | 12.5% |
| MegaCRN | 0.698 | 0.520 | 0.734 | 0.535 | 0.544 | 0.595 | 0.492 | 0.866 | 0.621 | 0.426 |
| MegaCRN+ | 0.667 | 0.591 | 0.684 | 0.600 | 0.521 | 0.666 | 0.458 | 0.907 | 0.590 | 0.477 |
| Improv | 4.44% | 13.7% | 6.81% | 12.1% | 4.23% | 11.9% | 6.91% | 4.73% | 5.00% | 12.0% |

Table 3: Model comparison for 7-day weather forecasting for Ningxia dataset.

| Model | 100m wind(U) | | 10m wind(U) | | Temperature | | Surface pressure | | Average Factor | |
|---|---|---|---|---|---|---|---|---|---|---|
| | RMSE↓ | ACC↑ | RMSE↓ | ACC↑ | RMSE↓ | ACC↑ | RMSE↓ | ACC↑ | RMSE↓ | ACC↑ |
| NWP | 0.968 | 0.521 | 0.933 | 0.514 | **0.319** | **0.844** | 0.325 | 0.961 | 0.901 | **0.526** |
| PINN | **0.697** | 0.470 | **0.681** | 0.437 | 0.635 | 0.654 | 0.494 | 0.904 | 0.666 | 0.387 |
| Bi-LSTM-T | 0.822 | 0.502 | 0.804 | 0.485 | 0.583 | 0.525 | 0.160 | 0.961 | 0.638 | 0.411 |
| Bi-LSTM-T+ | 0.798 | **0.545** | 0.777 | 0.520 | 0.560 | 0.584 | 0.156 | 0.965 | 0.616 | 0.445 |
| Improv | 2.92% | 8.57% | 3.36% | 7.22% | 4.28% | 11.2% | 2.50% | 0.42% | 3.45% | 8.27% |
| Hybrid-CBA | 0.842 | 0.456 | 0.819 | 0.445 | 0.652 | 0.430 | 0.150 | 0.964 | 0.657 | 0.338 |
| Hybrid-CBA+ | 0.801 | 0.536 | 0.790 | 0.509 | 0.563 | 0.589 | **0.149** | **0.966** | 0.621 | 0.416 |
| Improv | 4.87% | 17.5% | 3.54% | 14.4% | 13.7% | 37.0% | 0.67% | 0.21% | 5.48% | 18.8% |
| ConvLSTM | 0.865 | 0.429 | 0.848 | 0.408 | 0.592 | 0.499 | 0.175 | 0.959 | 0.656 | 0.364 |
| ConvLSTM+ | 0.826 | 0.477 | 0.814 | 0.472 | 0.520 | 0.619 | 0.170 | 0.955 | 0.622 | 0.419 |
| Improv | 4.51% | 11.2% | 4.01% | 15.7% | 12.2% | 24.0% | 2.86% | -0.42% | 5.18% | 15.1% |
| AFNO | 0.856 | 0.436 | 0.838 | 0.421 | 0.501 | 0.571 | 0.153 | 0.962 | 0.619 | 0.395 |
| AFNO+ | 0.823 | 0.505 | 0.808 | 0.498 | 0.466 | 0.693 | 0.153 | 0.956 | 0.596 | 0.456 |
| Improv | 3.86% | 15.8% | 3.58% | 18.3% | 6.99% | 17.9% | 0.00% | -0.31% | 3.72% | 15.4% |
| MTGNN | 0.835 | 0.484 | 0.820 | 0.465 | 0.502 | 0.526 | 0.162 | 0.958 | 0.617 | 0.395 |
| MTGNN+ | 0.810 | 0.525 | 0.792 | **0.521** | 0.469 | 0.677 | 0.160 | 0.959 | **0.595** | 0.455 |
| Improv | 2.99% | 8.47% | 3.41% | 12.0% | 6.57% | 28.7% | 1.96% | 0.10% | 3.57% | 15.2% |
| MegaCRN | 0.840 | 0.455 | 0.824 | 0.432 | 0.646 | 0.487 | 0.188 | 0.958 | 0.661 | 0.370 |
| MegaCRN+ | 0.809 | 0.510 | 0.793 | 0.485 | 0.598 | 0.600 | 0.183 | 0.954 | 0.629 | 0.432 |
| Improv | 4.64% | 12.1% | 3.76% | 12.3% | 7.43% | 23.2% | 2.66% | -0.42% | 4.84% | 16.8% |

each grid in both datasets, we select the few most important observational weather information for evaluation. All the weather factors are normalized during the preprocessing. More details of datasets are given in Appendix A.1. There are three categories of baseline models in comparison, including: (1) Meteorological models: Bi-LSTM-T (Yang et al., 2022), Hybrid-CBA (Han et al., 2022); (2) Vision models: ConvLSTM (Shi et al., 2015), AFNO (Guibas et al., 2021; Pathak et al., 2022); (3) Spatio-temporal graph models: MTGNN (Wu et al., 2020), MegaCRN (Jiang et al., 2023). Some of these baseline models are slightly modified to adapt to the multistep prediction setting, with details described in the Appendix A.2. Besides deep learning models, we also compare these models with the Numerical Weather Prediction (NWP) results provided by ECMWF and the Physical-Informed Neural Network (PINN) (Raissi et al., 2019) based on the PDEs discovered by *PhyDL-NWP*. We denote the baseline models as **BaseModels** and incorporate them with *PhyDL-NWP* as **BaseModels+**. We divide each dataset into train, validation, and test sets using an 8:1:1 ratio in chronological order. We perform multiple experiments based on the length of future prediction, ranging from one hour to seven days. Due to the GPU memory limitation, we use the eight weather factors of only ten hours in the past to predict all the eight weather factors in the future.

In particular, the result details with seven days are reported in Tables 2-3, as it represents the model's capability of long-term medium-range weather prediction, which is considered one of the biggest

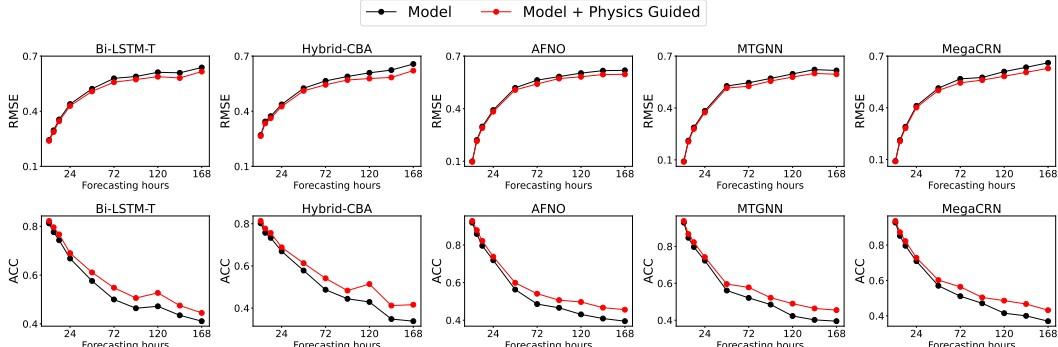

Figure 3: Model comparison in Ningxia dataset before and after physics guidance (+) for forecasting ranges from 1 hour to 7 days on the Average Factor.

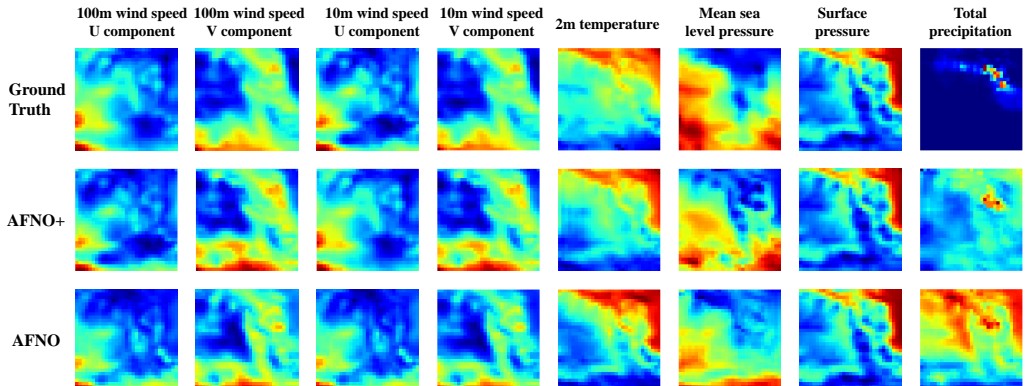

Figure 4: Example of comparison of 7-day weather forecast results with AFNO and AFNO+. The results of AFNO+ (guided by our *PhyDL-NWP* framework) are closer to the ground truth.

challenges in the field. The vanilla PINN does not seem to be effective, while the improvement provided by *PhyDL-NWP* is consistently significant. For Ningbo dataset, the overall improvement on Average Factor is 5.00% to 7.18% over RMSE and 9.48% to 14.0% over ACC; for Ningxia dataset, the overall improvement on that is 3.45% to 5.48% over RMSE and 8.27% to 18.8% over ACC. Moreover, it makes sense that the effectiveness on surface pressure seems trivial – surface pressure usually relates to the altitude, which is not included in our data. Furthermore, we find that the result of NWP is good in ACC which reflects the overall correlation and consistency of prediction provided by physical equations, while being the worst in RMSE which reflects the accuracy. Deep learning models, on the other hand, greatly outperform NWP in RMSE, showing great advantage in modeling capacity.

To understand the holistic properties of *PhyDL-NWP*, we conduct detailed analyses on the Ningxia dataset. First, the comparison of different models for different forecasting ranges on the Average Factor is visualized in Fig. 3. The comparison on the 100m wind component (U) is shown in Fig. 5. The comparisons on more weather factors are shown in Figs. 6-8 in Appendix A.4. Based on all the curves, **BaseModels+** always excels **BaseModels** and NWP at all time steps, which demonstrates that the *PhyDL-NWP* framework is consistently effective. As the forecasting range increases, the deep learning performance decreases significantly due to the generalization challenge. The improvement provided by *PhyDL-NWP*, however, is increasing in the forecasting range, which highlights its unique advantages of guiding models for long-term medium-range forecasting. In addition, based on one state-of-the-art model AFNO, we visualize the forecasting results of its vanilla version (AFNO) and the version guided by our *PhyDL-NWP* framework (AFNO+) in Fig. 4. It can be seen that *PhyDL-NWP* can improve the performance of the existing deep model, making the forecasting results closer to the ground truth.

### 4.3 PHYSICAL DISCOVERY

We fuse a priori terms in the NWP physical equations into our physical discovery. Upon the discovered PDEs from the three datasets, we find many cases that verify the effectiveness of *PhyDL-NWP* in not only representing but also improving the physical mechanisms for the weather data, which is generally consistent with the meteorology theory. For example, one equation that describes temperature evolution in the atmospheric boundary layer has the form:

$$\frac{\partial T}{\partial t} = -U\frac{\partial T}{\partial x} - V\frac{\partial T}{\partial y} - \omega\frac{\partial T}{\partial z} + k\frac{\partial^2 T}{\partial z^2} + H, \tag{10}$$

where $T$ denotes the temperature, $t$ denotes time, $x$ and $y$ denote space, $U$ and $V$ denote the horizontal wind components in the $x$ and $y$ directions, $z$ denotes the vertical height, $k$ denotes the eddy diffusivity for heat, and $H$ denotes various heating sources such as radiation. In contrast, PhyDL-NWP reflects physical mechanisms containing key terms above if possible. For Ningxia dataset,

$$\frac{\partial T}{\partial t} = -1.68U_{10}\frac{\partial T}{\partial x} - 1.59V_{10}\frac{\partial T}{\partial y} - 0.73U_{100}\frac{\partial T}{\partial x} - 0.68U_{100}\frac{\partial T}{\partial y} + ... + Q(x,y,t), \tag{11}$$

where ... contains other six terms that are less relevant in the PDE and $Q$ denotes the latent force. Since the vertical axis and the heat source are not included in the dataset, we assume that $Q(x,y,t)$ can capture these terms. Although $Q(x,y,t)$ is hard to interpret, from the performance of **BaseModels+**, we believe that the PDE does provide globally consistent guidance to help improve generalization. The less related terms might represent a way of correction to the major terms, since the key terms may be noisy in the dataset and not always informative. Moreover, the discovered physical mechanism is also consistent throughout different datasets. For Ningbo and Huadong datasets, we get similarly

$$\frac{\partial T}{\partial t} = -2.26U_{10}\frac{\partial T}{\partial x} - 2.03V_{10}\frac{\partial T}{\partial y} - 1.57U_{100}\frac{\partial T}{\partial x} - 1.26U_{100}\frac{\partial T}{\partial y} + ... + Q(x,y,t), \tag{12}$$

and

$$\frac{\partial T}{\partial t} = -1.65U_{10}\frac{\partial T}{\partial x} - 1.52V_{10}\frac{\partial T}{\partial y} - 0.79U_{100}\frac{\partial T}{\partial x} - 0.79U_{100}\frac{\partial T}{\partial y} + ... + Q(x,y,t). \tag{13}$$

Furthermore, the discovered mechanism for other weather factors also always captures the main terms in the theoretical PDEs. For example, the wind component has the following form:

$$\frac{\partial U}{\partial t} = -U\frac{\partial U}{\partial x} - V\frac{\partial U}{\partial y} - W\frac{\partial U}{\partial z} + \nu(\frac{\partial^2 U}{\partial x^2} + \frac{\partial^2 V}{\partial x^2} + \frac{\partial^2 W}{\partial x^2}) + F_{fx} \tag{14}$$

where $\nu$ means the kinematic viscosity and $F_{fx}$ represents the effect of friction (e.g., the difference between wind speeds at different heights, the roughness of the surface). These terms, not included in this case, usually need model parametrization. We can also discover similar forms of

$$\frac{\partial U_{10}}{\partial t} = -U_{10}\frac{\partial U_{10}}{\partial x} - V_{10}\frac{\partial U_{10}}{\partial y} - U_{100}\frac{\partial U_{10}}{\partial x} - V_{100}\frac{\partial U_{10}}{\partial y} + ... + Q(x,y,t), \tag{15}$$

where ... contains $\nabla P_{sea}$ and $U_{10}\nabla P_{sea}$, which describe how $\frac{\partial U_{10}}{\partial t}$ varies through advection and pressure gradient forces, since wind acceleration can be due to the pressure gradient force. Therefore, the less relevant terms in "..." may also convey some physical insights.

## 5 CONCLUSION

In this paper, we introduce a novel physics-guided approach for numerical weather forecasting named PhyDL-NWP. This approach combines the strengths of physics and deep learning to capture the nonlinear dynamics of meteorology, improving generalization in weather prediction tasks. PhyDL-NWP serves as a plug-and-play module aligning deep learning models with physical consistency for both medium-range weather forecasting and weather downscaling with unlimited granularity. The approach is validated through extensive experiments on three real-world datasets, showing significant performance improvement for state-of-the-art models under various circumstances. It also provides insights into understanding the underlying physical mechanisms of meteorology.

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

# A APPENDIX

## A.1 DATASET DESCRIPTION

For weather downscaling, the Huadong dataset consists of HRES and ERAs datasets. HRES represents a 10-day atmospheric model forecast, while ERA5 serves as a global atmospheric reanalysis, incorporating climate and weather observations. For regional downscaling, we construct a real-world dataset called "Huadong", covering the east China land and sea areas. In this dataset, HRES data is employed as the predictive data, while ERA5 reanalysis data serves as the ground truth.

**Huadong dataset**: The Huadong dataset encompasses a latitude range from $26.8°N$ to $42.9°N$ and a longitude range from $112.6°E$ to $123.7°E$. It comprises a grid of $64 \times 44$ cells, with each cell having a grid size of 0.25 degrees in both latitude and longitude. Notably, the Huadong dataset also incorporates Digital Elevation Model (DEM) data to represent terrain information. Since the terrain information usually refers to the boundary layers in the meteorology model instead of an individual weather factor in the PDE, for simplicity, we do not use this information in the paper. The HRES and ERA5 data cover the period from January 3, 2020, to April 1, 2022. The detailed weather factor descriptions of **Huadong** datasets are in Table 4. The scores of the average factors reported in Table 1 are computed based on all eight factors. Due to space limits, we only report the specific scores for four factors.

For weather forecasting, both Ningbo and Ningxia datasets consist of two main components: geographic data and meteorological data. The geographic data includes latitude, longitude, and DEM (Digital Elevation Model) information. The DEM information is commonly used in geographic information systems to represent the terrain of the area. On the other hand, the meteorological data in these datasets consist of various weather factors. These factors typically include wind speed, temperature, and pressure. These data provide information about the atmospheric conditions at different locations within the study area. To organize and represent the data, a grid format is used. In this format, the study area is divided into grids, and each grid cell represents a specific location. Within each grid cell, both the geographic and meteorological data for that location are stored.

**Ningbo dataset**: The Ningbo dataset represents a coastal area spanning from latitude 28.85°N to 30.56°N and longitude 120.91°E to 122.29°E. It is divided into a grid system comprising 58 grids in the latitude direction and 47 in the longitude direction. Each grid has a size of 0.03 degrees in both latitude and longitude. The DEM data are collected from ETOPO1[5]. The meteorological data are collected from Ningbo Meteorological Bureau[6], including 10 weather factors from 1/Jan/2021 to 1/Apr/2021 with 1-hour sample rate.

**Ningxia dataset**: The Ningxia dataset represents a mountainous area spanning from latitude 34.5°N to 42°N and longitude 106°E to 116°E. There are $30 \times 40$ grids with a grid size of 0.25 degrees in both latitude and longitude. The DEM data are collected from ETOPO1. The meteorological data are collected from ECMWF's ERA5[7], including 8 weather factors from 1/Jan/2021 to 1/Dec/2021 with 1-hour sample rate.

The statistics and weather factor descriptions of **Ningbo** and **Ningxia** datasets are in Table 5 and Table 6, respectively. In the forecasting experiments, we divide each dataset into train, validation, and test sets using an 8:1:1 ratio in chronological order. The scores of the average factors reported in Table 3 are computed based on all eight factors. Due to space limits, we only report the specific scores for four factors.

## A.2 BASELINES

### A.2.1 FORECASTING BASELINES

- Bi-LSTM-T Yang et al. (2022): a deep model that uses Bi-LSTM for weather prediction.

- Hybrid-CBA Han et al. (2022): a hybrid deep learning model that combines CNN, LSTM, and attention models for weather forecasting and correction.

---

[5]ETOPO1
[6]Ningbo Meteorological Bureau
[7]ECMWF's ERA5

Table 4: The statistics and weather factor descriptions of the Huadong dataset

| Statistics | |
|---|---|
| Time range | 2020.1.3 00:00-2022.4.1 24:00 |
| Temporal resolution | 1-hour |
| Number of time steps | 19368 |
| Latitude and longitude range | 26.8°N-42.9°N, 112.6°E-123.7°E |
| Spatial resolution | Origin: 0.25°, 2× : 0.5°, 4× : 1.0° |
| Number of grids | Origin: $64 \times 44$, $2\times : 32 \times 22$, $4\times : 16 \times 11$, |
| **Weather Factors** | |
| 100m U wind component | the horizontal speed of air moving towards the east, at a height of 100 meters above the surface of the Earth |
| 100m V wind component | the horizontal speed of air moving towards the north, at a height of 100 meters above the surface of the Earth |
| 10m U wind component | the horizontal speed of air moving towards the east, at a height of 10 meters above the surface of the Earth |
| 10m V wind component | the horizontal speed of air moving towards the north, at a height of 10 meters above the surface of the Earth |
| 2m temperature | the temperature of air at 2m above the surface of land, sea, or inland waters |
| Mean sea level pressure | the pressure of the atmosphere at the surface of the Earth, adjusted to the height of mean sea level |
| Surface pressure | the pressure of the atmosphere at the surface of land, sea, and inland water |
| Total precipitation | the accumulated liquid and frozen water, comprising rain and snow, that falls to the Earth's surface |

Table 5: The statistics and weather factor descriptions of the Ningbo dataset

| Statistics | |
|---|---|
| Time range | 2021.1.1 20:00-2021.4.1 20:00 |
| Temporal resolution | 1-hour |
| Number of time steps | 2880 |
| Latitude and longitude range | 28.85°N-30.56°N, 120.91°E-122.29°E |
| Spatial resolution | 0.03 degree |
| Number of grids | 2726 |
| **Weather factors** | |
| 2m temperature | the temperature of air at 2 meters above the surface of land, sea, or inland waters |
| Total precipitation | the accumulated liquid and frozen water, comprising rain and snow, that falls to the Earth's surface |
| 2m relative humidity | the measure of the amount of moisture or water vapor present in the air compared to the maximum amount of moisture that the air could hold at a specific temperature, at 2 meters above the surface of land, sea, or inland waters |
| Pressure | the pressure of the atmosphere at the surface of land, sea, and inland water |
| 10m relative vorticity | the rotation of the air at a height of 10 meters above the Earth's surface, relative to the Earth's rotation |
| 10m divergence | the measure of the expansion or spreading out of air at a height of 10 meters above the Earth's surface |
| 10m wind speed | the speed at which the wind is blowing at a height of 10 meters above the Earth's surface |
| 10m wind direction | the direction from which the wind is coming at a height of 10 meters above the Earth's surface |
| 100m wind speed | the speed at which the wind is blowing at a height of 100 meters above the Earth's surface |
| 100m wind direction | the direction from which the wind is coming at a height of 10 meters above the Earth's surface |

- ConvLSTM Shi et al. (2015): a hybrid deep learning model that extends LSTM with convolutional gates.

- AFNO Guibas et al. (2021); Pathak et al. (2022): a deep learning model that adapts Fourier neural operator for spatio-temporal modeling.

- MTGNN Wu et al. (2020): a deep learning model that learns multivariate time series with graph neural networks.

Table 6: The statistics and weather factor descriptions of the Ningxia dataset

| | |
|---|---|
| Time range | 2021.1.1 0:00-2022.1.1 0:00 |
| Temporal resolution | 1-hour |
| Number of time steps | 8760 |
| Latitude and longitude range | 34.5°N-42°N, 106°E-116°E |
| Spatial resolution | 0.25 degree |
| Number of grids | 1271 |
| Weather factors | |
| 100m U wind component | the horizontal speed of air moving towards the east, at a height of 100 meters above the surface of the Earth |
| 100m V wind component | the horizontal speed of air moving towards the north, at a height of 100 meters above the surface of the Earth |
| 10m U wind component | the horizontal speed of air moving towards the east, at a height of 10 meters above the surface of the Earth |
| 10m V wind component | the horizontal speed of air moving towards the north, at a height of 10 meters above the surface of the Earth |
| 2m temperature | the temperature of air at 2m above the surface of land, sea, or inland waters |
| Mean sea level pressure | the pressure of the atmosphere at the surface of the Earth, adjusted to the height of mean sea level |
| Surface pressure | the pressure of the atmosphere at the surface of land, sea, and inland water |
| Total precipitation | the accumulated liquid and frozen water, comprising rain and snow, that falls to the Earth's surface |

- MegaCRN Jiang et al. (2023): a deep learning model that learns heterogeneous spatial relationships with adaptive graphs.

Since most forecasting baselines are designed for single-step future prediction by default, we modify their neural architecture by multiplying the output dimension of the second-last layer (usually at the end of an LSTM or Conv block, before passing through the feed-forward network at the end) by the number of prediction steps.

### A.2.2 DOWNSCALING BASELINES

- Bicubic interpolation Keys (1981): a two-dimensional interpolation technique that uses the values and gradients of the function at surrounding grid points to obtain a smooth and continuous interpolated result.

- FSRCNN Passarella et al. (2022): a widely recognized method in computer vision, leveraged for both downscaling and single-image super-resolution, which conducts feature mapping using multi-layer CNNs and executes upsampling via deconvolution layers.

- ResDeepD Sharma & Mitra (2022): a deep model that begins with an upsampling of the input to increase dimensions before proceeding to feature mapping via ResNet.

- EDSR Jiang & Chen (2022): a deep model that first conducts feature mapping using ResNet and then performs upsampling.

- RCAN Yu et al. (2021b): a deep model based on ResNet that incorporates a global pooling layer for channel attention.

- YNet Liu et al. (2020): a novel deep convolutional neural network (CNN) with skip connections and fusion capabilities to perform downscaling for climate variables.

- DeepSD Vandal et al. (2017): a generalized stacked super resolution convolutional neural network (SRCNN) framework for statistical downscaling of climate variables.

### A.3 MODEL PARAMETERS

The size of the parameters of different models is summarized in Table 7.

Table 7: The comparison of model parameters for 7-day weather forecasting.

| Model | PhyDL-NWP | BiLSTM | Hybrid-CBA | ConvLSTM | AFNO | MTGNN | MegaCRN |
|---|---|---|---|---|---|---|---|
| #Parameters | 54984 | 171M | 198M | 678272 | 520512 | 1660960 | 580176 |

## A.4 MORE MODEL COMPARISON

Here we plot more results on model comparison before and after physics guidance for forecasting ranges from 1 hour to 7 days on different circumstances in Table 6 (on the 10m wind (U)), Table 7(on the temperature), and Table 8(on the surface pressure). Overall, the proposed physics guidance brings performance improvement in all cases.

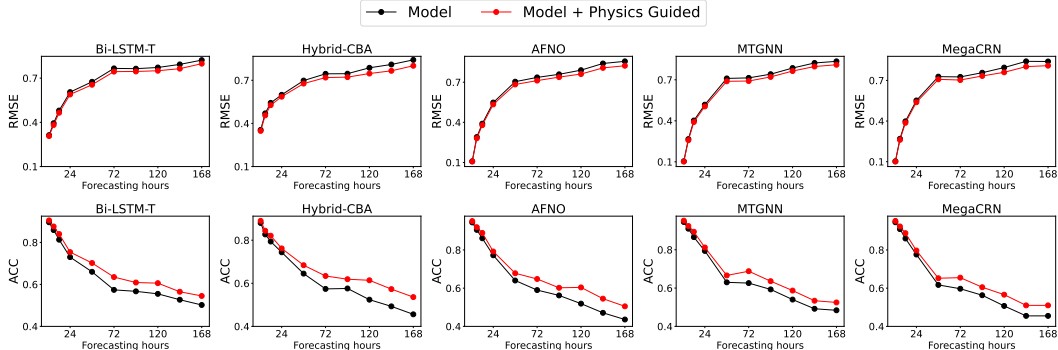

Figure 5: Model comparison in Ningxia dataset before and after physics guidance (+) for forecasting ranges from 1 hour to 7 days on the 100m wind (U).

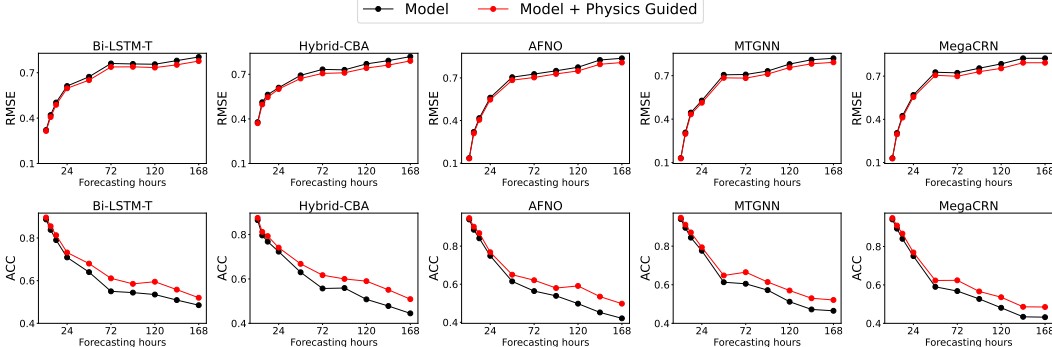

Figure 6: Model comparison in Ningxia dataset before and after physics guidance (+) for forecasting ranges from 1 hour to 7 days on the 10m wind (U).

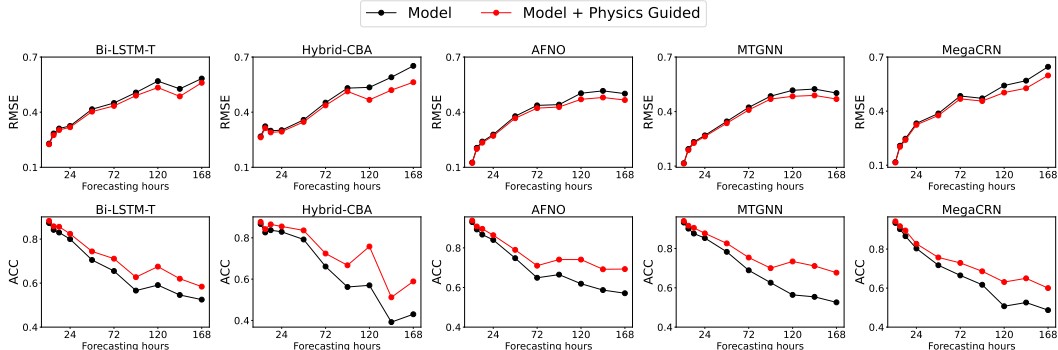

Figure 7: Model comparison in Ningxia dataset before and after physics guidance (+) for forecasting ranges from 1 hour to 7 days on the temperature.

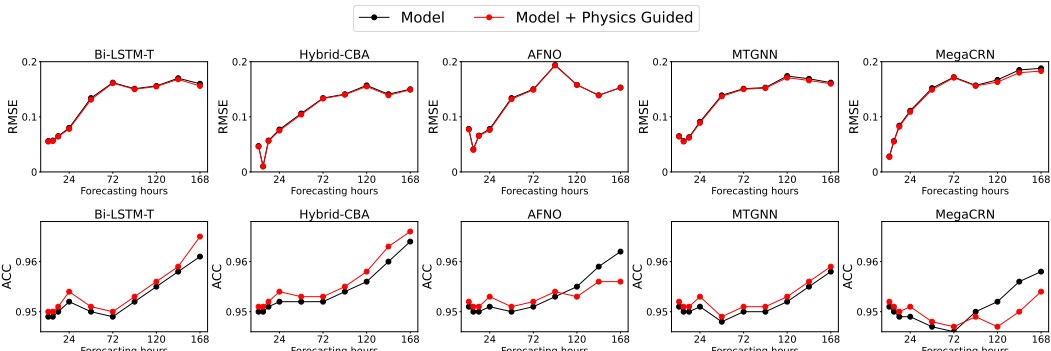

Figure 8: Model comparison in Ningxia dataset before and after physics guidance (+) for forecasting ranges from 1 hour to 7 days on the surface pressure.

