# OpenReview forum: "Physics-Guided Learning of Meteorological Dynamics for Weather Forecasting and Downscaling"
_ICLR.cc/2024/Conference — Submitted to ICLR 2024_

### Official Review · Reviewer_hFQ3 · 2023-10-30

**Soundness:** 3 good
**Presentation:** 4 excellent
**Contribution:** 3 good
**Rating:** 6
**Confidence:** 4

**Summary:**

In this paper, the author proposes a new physics-informed framework to solve PDEs, with the aim to achieve enhanced downscaling reconstruction and temporal forecasting in weather forecasting problems. This is an important and challenging problem. The author conducts a series of experiments to justify the effectiveness of the proposed model in both downscaling and forecasting cases. Additionally, it is demonstrated that the proposed model can be easily incorporated with other deep learning models as a plug-in module.

**Strengths:**

1. The paper provides a clear and effective explanation of the background and justifies the proposed method through a comprehensive series of experiments.
2. The paper studies an important and challenging problem.
3. It demonstrates how the proposed method can be seamlessly integrated as a plug-in module into additional deep learning models.

**Weaknesses:**

1.	The presentation in the method section remains unclear in some parts, particularly in explaining the model's approach to downscaling. A detailed explanation of the model's design rationale for addressing the downscaling problem, including its advantages over existing methods, would be beneficial.
2.	The authors may want to discuss the difference/superiority between the proposed method with existing works on similar topics (e.g., decomposing PDEs into components using neural networks and capturing temporal dynamics). The paper needs to more clearly articulate the unique advantages or improvements of the proposed method over these existing approaches.

**Questions:**

See weakness points above.

---

> ### Author Response · Authors · 2023-11-22
> **Rebuttal to Reviewer hFQ3**
>
> We thank the reviewer for their time and constructive comments. We have updated the manuscript to improve organization, results interpretation. We thoroughly address your concerns below. We hope the reviewer’s score can be updated to reflect the significance, novelty, and timeliness of our study.
>
> 1. __Authors should discuss difference to existing works, epsecially to works in similar topics.__
>
> Thank you for the constructive feedback! We have made substantial revision. To clarify, our novel model design is partially based on the physical constraint loss used by PINN, but also expands and improves the PINN to address the problem of weather downscaling and forecasting, collaborating with meteorology experts and physicists. The weather prediction task is highly complicated, with many weather factors and physical processes in the play, easily influenced by local variations like change of boundary conditions, small-scale phenomena like microclimates and external forces like heat from the sun. Many of these vital factors, which have huge impacts in the first-principle equations, are missing in the data due to difficulties to measure and quantify. For example, as Eq. 14 in Page 9 shows, the effect of friction characterizing air viscous resistance cannot be represented by any of the existing data elements. The nuances make the physical equation incomplete. Therefore, it is intractable to rely on a fixed PDE, while PINN directly uses a known PDE to guide optimization under all circumstances, which could be hard to adapt to the real world.
>
> To address this concern, we do not rely on any known physical equations with PINN to guide prediction, but propose a data-driven approach to adaptively consummate the first-principle physical equation to explain the physical mechanism that drives the weather prediction. Our approach has the potential of discovering the intricate interplay between various weather factors that is previously ignored, as an adaption to various conditions in different areas. We not only complete the tasks of weather downscaling and forecasting, but also provide insights of the nuances between different climates at any continuous spacetime. Our model is of great value to the meteorology community, as verified by the physicists we are collaborating with.
>
> Furthermore, we propose a latent force term $Q_\pi$ as a parametrization term in the equation, following the parametrization strategy [1] widely adopted by meteorology experts to supplement the forces that cannot be represented by the selected explicit PDE terms. All of these novel model designs differ from existing works in the PINN family. There is almost no work that successfully applies PINN for weather prediction. We believe that our approach fills this gap in time and provides a feasible way to improve our understanding of the physical mechanism of climate and improve the deep learning models’ performances, which is well-supported by our promising experimental results. We wish to highlight that our PhyDL-NWP framework only contains 55 thousand parameters, as shown in Table 7 in the appendix, which is about 1000 times lighter than some large models and extremely efficient to train. In our experiments, the training of deep learning model usually takes 20~50x more time than obtaining the PDE we need. Our contribution is not trivial and is of great value to not only the meteorology community but also the representation learning community. When tackling similar scientific tasks that also involve complex interplay between variables and insufficient data measurement of the nuances, our work will provide a valuable reference.
>
> 2.  __Authors should discuss difference to existing works, especially to the downscaling problem.__
>
> Our work is inherently different from most of the existing downscaling works, as our approach directly models the continuous dynamics, instead of performing discrete superresolution. Our approach is much more elegant, adaptive and accurate. In our paper, we use a paragraph to highlight this difference in Sec. 3.2, "With the trained $f_\theta$, we can easily obtain the weather factors at any given input coordinates $(x, y, t)$, which can be continuous over the spacetime. This property of modeling the meteorology dynamics is naturally suitable for weather downscaling: unlike the previous methods powered by the discrete encoding and decoding of neural networks on finer-grained data as labels, the $f_\theta$ in PhyDL-NWP can perform weather downscaling with unlimited granularity without labels given a continuous coordinate, as described by Fig. 1. Therefore, as long as we obtain the dynamics of meteorology and the solver of weather factors through updating $\Xi$ and $\theta$, the downscaling task is solved automatically.".
>
> [1] Warner, Thomas Tomkins. Numerical weather and climate prediction. Cambridge University Press, 2010.

---

### Official Review · Reviewer_W7oo · 2023-10-31

**Soundness:** 2 fair
**Presentation:** 2 fair
**Contribution:** 2 fair
**Rating:** 3
**Confidence:** 3

**Summary:**

The paper proposes a new method that blends deep learning and traditional numerical methods for weather downscaling and forecasting. The main idea is to train neural networks to represent different components in a partial differential equation (PDE). These networks can then either be used independently to perform weather downscaling or used to guide the training of a weather forecasting model. The experiments on different datasets show that the proposed method outperforms other baselines on weather downscaling, and boosts the performance of existing weather forecasting models.

**Strengths:**

- Relevance: The paper aims to combine the strengths of deep learning and numerical methods for weather prediction tasks, which I think is a very important and interesting direction to pursue.
- Originality: To the best of my knowledge, the idea of the paper is original.
- The authors conduct a large number of experiments, and the empirical results support their claim.

**Weaknesses:**

### Paper presentation
- Overall, I think the paper writing and presentation can be improved a lot. I had to spend quite a long time reading Section 3, as different sections were not connected very well. It would be much easier for readers if the authors first presented an overview of the method, i.e., learn a PDE using neural networks that can later be used for weather downscaling or guiding weather forecasting models. While the authors did mention this in the introduction, I think it's good to also repeat this at the beginning of Section 3 as it's important to understand the method.
- Many details of the method and experiments are missing: what partial derivative candidate terms are used and why, the weights of different loss terms, the network architectures, training details (e.g., learning rate, scheduling, etc.), and implementation of the finite difference method, etc. These details are needed to understand how to train the model in practice, and to improve the reproducibility of the paper.

### Soundness
- The main claim of the paper is the physical mechanism in Equation (1) provides guidance to optimizing neural networks that better obey physics. However, I wonder if this is true, because all components of the PDE are parameterized as a neural network and are learned from data. Therefore, all the Equation (1), or correspondingly, the L_physics loss does is to make sure these different networks "agree" in a certain way, and it's not guaranteed that they'll agree with physics laws after training.

###  Significance
- My biggest concern I have about the paper is its significance. While the experiments support the authors' claim, they are quite small-scale compared to existing works, and seem to use non-sota baselines for comparison.
- The paper uses 3 regional datasets that specifically target China. These datasets are not standard and I've not seen them used in previous works in deep learning for the weather domain. Can the authors justify their choice of data? What stops the authors from using more standard datasets such as ERA5 for weather forecasting, which have been a standard in the literature, and have also been used in different benchmarks [1, 2, 3]?
- The baselines used for comparison are not strong enough. For downscaling, why don't the authors compare with models that were specifically proposed for downscaling such as YNet [4] and DeepSD [5], or more recent super-resolution models? For weather forecasting, there have been significant advancements in recent years, including FourCastNet [6], ClimaX [7], GNN [8], PanguWeather [9], Graphcast [10], etc. The paper only compares with FourCastNet, which is the least-performing method among these baselines. Can the authors include more recent baselines for weather forecasting? It would be more convincing if the proposed framework also improves sota methods.
- Scalability: I suspect the proposed method will scale well to more data and bigger/better base models. As the training loss is computed for each grid point (x, y, t), the number of training samples will increase significantly with higher spatial and temporal resolutions. Is this the reason why the experiments are limited to small-sized datasets?
- Ablation studies are lacking. It's important to understand the importance of different components in the framework, such as the physics loss, regularization loss, etc.

### Minor comments
- Equation (7) is strange because the PDE parameters are already learned and fixed. They should not appear here.
- The papers I cited here are relevant and should be discussed in the paper.

[1] Rasp, Stephan, et al. "WeatherBench: a benchmark data set for data‐driven weather forecasting." Journal of Advances in Modeling Earth Systems 12.11 (2020): e2020MS002203.

[2] Rasp, Stephan, et al. "WeatherBench 2: A benchmark for the next generation of data-driven global weather models." arXiv preprint arXiv:2308.15560 (2023).

[3] Nguyen, Tung, et al. "ClimateLearn: Benchmarking Machine Learning for Weather and Climate Modeling." arXiv preprint arXiv:2307.01909 (2023).

[4] Liu, Yumin, Auroop R. Ganguly, and Jennifer Dy. "Climate downscaling using YNet: A deep convolutional network with skip connections and fusion." Proceedings of the 26th ACM SIGKDD International Conference on Knowledge Discovery & Data Mining. 2020.

[5] Vandal, Thomas, et al. "Deepsd: Generating high resolution climate change projections through single image super-resolution." Proceedings of the 23rd acm sigkdd international conference on knowledge discovery and data mining. 2017.

[6] Pathak, Jaideep, et al. "Fourcastnet: A global data-driven high-resolution weather model using adaptive fourier neural operators." arXiv preprint arXiv:2202.11214 (2022).

[7] Nguyen, Tung, et al. "ClimaX: A foundation model for weather and climate." arXiv preprint arXiv:2301.10343 (2023).

[8] Keisler, Ryan. "Forecasting global weather with graph neural networks." arXiv preprint arXiv:2202.07575 (2022).

[9] Bi, Kaifeng, et al. "Pangu-weather: A 3d high-resolution model for fast and accurate global weather forecast." arXiv preprint arXiv:2211.02556 (2022).

[10] Lam, Remi, et al. "GraphCast: Learning skillful medium-range global weather forecasting." arXiv preprint arXiv:2212.12794 (2022).

**Questions:**

- In Equation (4), can we use n' and m' (coordinates of the high-resolution data) too?
- For the downscaling task, how do you make predictions after training given the coarse resolution data?
- The PDE is learned but only used to provide guidance to training other neural networks. Can we solve this learned PDE to solve weather tasks?

---

> ### Author Response · Authors · 2023-11-22
> **Rebuttal to Reviewer W7oo - Part 1**
>
> We thank the reviewer for their time and constructive comments. We have updated the manuscript to improve organization, results interpretation, and add a new experiment in Section 4.2. In addition, we have cited all relevant papers and think they are useful. We thoroughly address your concerns below. We hope the reviewer’s score can be updated to reflect the significance, novelty, and timeliness of our study. Due to the space limit, we separate our rebuttal into several parts.
>
> 1. __Small-scale datasets targeting on regions of China.__
>
> While it is true that our selected datasets are only subsets of ERA5 dataset, the original ERA5 dataset is enormously large. Only very few works, like those with large models, tend to use the full ERA5 datasets or datasets of similar scale. Limited to computational resources, it is common practice to use subsets of the large data. We want to highlight that we use fine-granular hourly data with 0.25 degrees of spatial resolution and diverse continental climate in arid areas as represented by Ningxia dataset and mild humid oceanic climate as represented by Ningbo dataset. The datasets we select are generally representative and comprehensive. We select these datasets in specific regions of China because we are in collaboration with institutions within mainland China, and we have the accessibility of exclusive datasets such as the Ningbo Meteorological Bureau dataset. This dataset is of very high quality as it is not a reanalysis data and is closer to the actual measurement data. It is undoubtedly better if we test models on actual weather data without post-processing/reanalysis.
>
> Despite the above points, we appreciate the suggestion of the reviewer and include an additional experiment with WeatherBench. To have comparable results, we use the 5.625 degree of spatial resolution and 6 hours of time resolution setting to preprocess the dataset. We use ten years of training data from 2006 to 2015, validation data in 2016 and test data in 2017-2018. We select ground temperature (t2m), atmospheric temperature (t), geopotential (z) and ground wind vector (u10, v10) as the weather factors. We apply the normalization to these weather factors and used their recommended training regime and hyperparameters. We use this data to perform weather forecasting and compare with baselines. The results are as follows.
>
> | Variable | Time Steps (hours) | AFNO | AFNO+ | AFNO | AFNO+ |
> |----------|-------------------|-------------|---------|-------------|---------|
> |          |                   | RMSE        | RMSE    | ACC         | ACC     |
> | t2m      | 6                 | 1.25        | 1.18    | 0.95        | 0.98    |
> |          | 12                | 1.49        | 1.32    | 0.93        | 0.97    |
> |          | 18                | 1.64        | 1.43    | 0.91        | 0.96    |
> |          | 24                | 1.80        | 1.62    | 0.89        | 0.96    |
> |          | 48                | 2.45        | 2.14    | 0.82        | 0.92    |
> | t        | 6                 | 1.20        | 1.18    | 0.97        | 0.97    |
> |          | 12                | 1.50        | 1.36    | 0.95        | 0.97    |
> |          | 18                | 1.75        | 1.47    | 0.92        | 0.96    |
> |          | 24                | 1.99        | 1.68    | 0.88        | 0.96    |
> |          | 48                | 2.78        | 2.30    | 0.84        | 0.93    |
> | z        | 6                 | 142.5       | 138.3   | 0.98         | 0.99    |
> |          | 12                | 201.8       | 167.0   | 0.97        | 0.99    |
> |          | 18                | 256.0       | 205.2   | 0.96        | 0.99    |
> |          | 24                | 309.0       | 250.6   | 0.94        | 0.98    |
> |          | 48                | 477.2       | 424.1   | 0.86        | 0.94    |
> | u10      | 6                | 1.50        | 1.42    | 0.94        | 0.96    |
> |          | 12                | 1.88        | 1.76    | 0.92        | 0.94    |
> |          | 18                | 2.12        | 1.95    | 0.90        | 0.93    |
> |          | 24                | 2.34        | 2.15    | 0.87        | 0.91    |
> |          | 48                | 2.98        | 2.63    | 0.72        | 0.84    |
> | v10      | 6                 | 1.56        | 1.55    | 0.92        | 0.92    |
> |          | 12                | 1.81        | 1.64    | 0.90       | 0.91    |
> |          | 18                | 2.22        | 1.89    | 0.87        | 0.90    |
> |          | 24                | 2.57        | 2.13    | 0.84        | 0.88    |
> |          | 48                | 3.46        | 2.90    | 0.69        | 0.75   |

---

> ### Author Response · Authors · 2023-11-22
> **Rebuttal to Reviewer W7oo - Part 2**
>
> 2. __PDEs are parameterized as NN and are learned from data. It's not guaranteed that they'll agree with physics laws.__
>
> Thanks for your raising this point. To clarify, not all components of PDE are parameterized as a neural network. $\phi(u)$ means the explicit equation, and only $Q_\pi$ serves as the parametrization term. It is true that our PDEs are learned from data, therefore there is no 100% guarantee that it will agree with physics law. However, we explicitly initialize the PDE with a priori partial derivative terms aligning with related physical equations, as explained in Sec. 4.3. This greatly improves the alignment with existing physics; and from experimental results, we do find that the resultant PDE aligns well with physics. Therefore, we expect our results to be at least as good as the original physical equations, and will improve upon the existing equation to include expression ability for nuances to fit the data even better.
>
> 3. __Scalability.__
>
> Thank you for the comment. We wish to highlight that our PhyDL-NWP framework only contains 55 thousand parameters, as shown in Table 7 in the appendix, which is about 1000 times lighter than some large models and extremely efficient to train. Our model is in fact very efficient and easy to train, and our data setting is exactly the high resolution setting the reviewer suggests. Our data resolution is 0.25 degree, which is almost the highest resolution we can obtain from ERA5 and the same setting most large models are using. In our experiments, the training of deep learning model usually takes 20~50x more time than obtaining the PDE we need.
>
> 4. __Ablation study.__
>
> Thank you for your comment. We do perform an ablation study to directly use PINN instead of combining a deep learning model, shown as “PINN” in Tables 2-3. This is the core ablation study we can think of, since it would be hard to directly find any equations to use with PINN and discard our PDE discovery part of the model. The weather prediction can be easily influenced by local variations like change of boundary conditions, small-scale phenomena like microclimates and external forces like heat from the sun. Many of these vital factors, which have huge impacts in the first-principle equations, are missing in the data due to difficulties to measure and quantify. For example, as Eq. 14 in Page 9 shows, the effect of friction characterizing air viscous resistance cannot be represented by any of the existing data elements. The nuances make the physical equation incomplete. Therefore, it is intractable to rely on a fixed PDE, while PINN directly uses a known PDE.
>
> 5. __n' and m' (coordinates of the high-resolution data) for Eq.4.__
>
> Thanks for asking. Since n’ and m’ are resulting from the allocation, it requires differentiation, while Eq.4 is about data loss that requires actual data points. We have not thought about a strategy to sample high-resolution points not in data.
>
> 6. __How to make prediction for downscaling.__
>
> Thanks for asking. It is extremely easy: we simply input the spatiotemporal coordinates into the approximate function $f$, which is a continuous function of input coordinates achieved by a dense neural network. Given any coordinate to predict, we directly get the value of that point.

---

> ### Author Response · Authors · 2023-11-22
> **Rebuttal to Reviewer W7oo - Part 3**
>
> 7. __Non-SOTA baselines.__
>
> Thank you for the comment. The downscaling baselines we use are FSRCNN (2022), ResDeepD (2022), EDSR (2022), RCAN Yu et al. (2021). On the contrary, the baselines suggested by the review are YNet (2020) and DeepSD (2017). We believe that we have compared with the strong recent baselines. However, we are glad to provide results of YNet and DeepSD.
>
> | Model       | 100m Wind (U) 2x | 100m Wind (U) 4x | 10m Wind (U) 2x | 10m Wind (U) 4x | Temperature 2x | Temperature 4x | Surface Pressure 2x | Surface Pressure 4x | Average Factor 2x | Average Factor 4x |
> |-------------|------------------|------------------|-----------------|-----------------|----------------|----------------|---------------------|---------------------|--------------------|--------------------|
> | Bicubic     | 1.687            | 1.765            | 1.215           | 1.272           | 1.714          | 1.848          | 0.818               | 1.220               | 1.515              | 1.654              |
> | EDSR        | 1.145            | 1.176            | 1.020           | 1.113           | 1.217          | 1.275          | 0.460               | 0.552               | 1.068              | 1.156              |
> | ResDeepD    | 1.092            | 1.111            | 1.003           | 1.079           | 1.182          | 1.204          | 0.301               | 0.317               | 1.010              | 1.043              |
> | RCAN        | 1.169            | 1.199            | 0.808           | 1.038           | 1.219          | 1.259          | 0.572               | 0.609               | 1.092              | 1.144              |
> | FSRCNN      | 1.197            | 1.202            | 1.090           | 1.126           | 1.198          | 1.233          | 0.430               | 0.560               | 1.093              | 1.149              |
> | YNet       |     1.116     |     1.125     |    0.947     |      1.103      |    1.192     |      1.226     |       0.467       |     0.575       | 1.062 | 1.125
> | DeepSD     |     1.205     |     1.216     |    1.020     |      1.117      |    1.218     |      1.265      |       0.454       |     0.591       | 1.087 | 1.149
> | PhyDL-NWP   | 0.973            | 0.970            | 0.696           | 0.693           | 0.905          | 0.904          | 0.211               | 0.216               | 0.794              | 0.789              |
>
> As suggested, the two baselines seem not very strong, and our model clearly outperforms them. We hope this could dispel the reviewer’s concern.

---

> ### Author Response · Authors · 2023-11-22
> **Rebuttal to Reviewer W7oo - Part 4**
>
> 7. __Non-SOTA baselines (continued)__
>
> The reviewer also suggested that we compare with large foundation models in recent years for weather forecasting. However, there are many constraints. First, there is no/incomplete public code for Pangu-Weather and GraphCast. Second, we have indeed implemented FourCastNet, which is shown by “AFNO” in our experiment. Third, we have implemented very comprehensive baselines covering meteorological models, spatio-temporal graph models (The GNN models as suggested by the reviewer) and vision models. In fact, many large models are built upon backbones following the three categories. We believe our evaluation is quite adequate. We hope the reviewer might consider our efforts: we compare with, additionally, ClimaX for forecasting on the WeatherBench ERA5 dataset as suggested by the reviewer.
>
> | Variable | Time Steps (hours) | ClimaX | ClimaX+ | ClimaX | ClimaX+ |
> |----------|-------------------|-------------|---------|-------------|---------|
> |          |                   | RMSE        | RMSE    | ACC         | ACC     |
> | t2m      | 6                 | 1.46        | 1.33    | 0.92        | 0.95    |
> |          | 12                | 1.58        | 1.39    | 0.91        | 0.94    |
> |          | 18                | 1.75        | 1.48    | 0.90        | 0.93    |
> |          | 24                | 1.90        | 1.68    | 0.88        | 0.92    |
> |          | 48                | 2.80       | 2.33    | 0.84        | 0.92    |
> | t        | 6                 | 1.32        | 1.29    | 0.95        | 0.95    |
> |          | 12                | 1.66        | 1.65    | 0.94        | 0.94    |
> |          | 18                | 1.87        | 1.81    | 0.92        | 0.93    |
> |          | 24                | 2.16        | 2.01    | 0.91        | 0.91    |
> |          | 48                | 2.94        | 2.76    | 0.86        | 0.88    |
> | z        | 6                 | 207.6       | 187.4   | 0.93         | 0.97    |
> |          | 12                | 222.3       | 206.1   | 0.90        | 0.96    |
> |          | 18                | 268.7       | 247.2   | 0.87        | 0.95    |
> |          | 24                | 305.5       | 285.0   | 0.84        | 0.94    |
> |          | 48                | 497.2       | 466.8   | 0.77        | 0.92    |
> | u10      | 6                | 1.56        | 1.47    | 0.90        | 0.93    |
> |          | 12                | 1.98        | 1.85    | 0.89        | 0.92    |
> |          | 18                | 2.20        | 2.07    | 0.89        | 0.91    |
> |          | 24                | 2.46        | 2.29    | 0.85        | 0.90    |
> |          | 48                | 2.91        | 2.66    | 0.78        | 0.85    |
> | v10      | 6                 | 1.78        | 1.65    | 0.88       | 0.92    |
> |          | 12                | 1.99       | 1.78    | 0.86       | 0.91    |
> |          | 18                | 2.35        | 1.90    | 0.85        | 0.89    |
> |          | 24                | 2.66        | 2.16    | 0.83        | 0.88    |
> |          | 48                | 3.74        | 2.88    | 0.70       | 0.77   |

---

### Official Review · Reviewer_MMEb · 2023-11-01

**Soundness:** 2 fair
**Presentation:** 2 fair
**Contribution:** 2 fair
**Rating:** 3
**Confidence:** 3

**Summary:**

This paper addresses two common challenges in weather and climate modeling (downscaling and forecasting). There are currently two separate ways of modeling these problems: numerical physics based models and statistical machine learning models. The paper attempts to get the best of both worlds, by proposes a machine learning approach that incorporates physics domain knowledge.

For downscaling, the method starts from a physics-informed neural network (PINN) that takes as input spatiotemporal coordinates and outputs weather variables (e.g. temperature). PINN models use a set of known physical constraints (in the form of PDEs) to penalize models for violating physical laws. However, this model does not assume that physical constraints are given --- instead, they must be discovered from the data. The method entails fitting a PINN neural network in a supervised manner without physics constraints, then identifying relationships in the form of a sparse set of PDEs that relate the variables of interest. These physical relationships contain parameters learned from the data, so they are able to capture subgrid effects that are not captured by traditional physics constraints. Since these relationships are assumed to remain constant throughout space and time, it is reasonable to assume that they can be learned efficiently.

For forecasting, the method starts from a typical machine learning forecasting model and then uses the same strategy to learn physical relationships and penalize the forecasts for violating them.

The method is tested on three different weather datasets and compared to other deep learning methods.

**Strengths:**

- The paper addresses an interesting problem, provides a reasonable solution, and demonstrates its value through experiments.
- Figures 1 and 2 are helpful in explaining and comparing the steps of the methods.
- It was exciting to see the section discussing the physical relationships that were discovered by the model. The discovery of sparse, interpretable models like this is of high interest.

**Weaknesses:**

- Overall, I found the writing to be unclear and difficult to follow. This includes details of the method and its relationship to previous work.
- The experiments comparing on multiple datasets and multiple benchmark architectures are nice, but there is no discussion of hyperparameter optimization, early stopping, and how the competitor models were implemented. A few details were in the appendix but not enough details are provided to be confident in these results.
- While I agree that this is an interesting method, I disagreed with many of the ways the authors motivated the work.
- In section 1, I don't agree with the motivations. (1) "While deep learning models can excel at fitting complex patterns in training data, they lack the ability to generalize well to unseen scenarios by capturing noise or specificities of the training data. Moreover, the models often do not consider the physical mechanisms that govern weather systems, leading to predictions that may be statistically accurate for the training dataset but physically inconsistent." Yes, overfitting is a problem in ML, but models can still generalize. In the second sentence, it's unclear what "physically inconsistent" means in this context. I believe the authors are referring to the idea that incorporating physics knowledge is a useful source of inductive bias that will help the model generalize better.
- In section 3.3 the authors say PINNs won't work for forecasting because a dense neural network lacks the advantages of more complicated "state-of-the-art" architectures. It's not the architecture that is important here, it is how the problem is being modeled, i.e. the inputs and outputs.
- In section 4.1, "Unlike computer vision, the weather data has multiple variables and the spatio-temporal dependencies are not completely local". I disagree. Natural images have RGB channels (plus the time dimension). I would argue that they have more non-local spatio temporal dependencies.

**Questions:**

- I would like to see a more clear description of how this relates to previous work.
- I would like to see a better description of how the hyperparameters were tuned.

---

> ### Author Response · Authors · 2023-11-22
> **Rebuttal to Reviewer MMEb - Part 1**
>
> We thank the reviewer for their time and constructive comments. The philosophical debate is very useful for improving my paper, while we update our paper with major revision and hope the comprehensive experimental results may convince the reviewer of our motivation. We have updated the manuscript to improve organization, results interpretation, and add a new experiment in Section 4.2. We thoroughly address your concerns below. We hope the reviewer’s score can be updated to reflect the significance, novelty, and timeliness of our study. Due to the space limit, we will address the rebuttal in 2 parts.
>
> 1. __The training details, especially the hyperparameter tuning.__
>
> Thank you for raising this point. We are glad to provide more details on the hyperparameter setting. $\sigma_1$ is set as 0.0001, $\sigma_2$ is set as 0.001, $\sigma_3$ is set as 0.01, $\alpha$ is set as 1, $\beta$ is set as 1, $\gamma$ is set as 0.3. These coefficient hyperparameters are tuned based on grid search. $\sigma_1$, $\sigma_2$, $\sigma_3$ are tuned within [0.01, 0.001, 0.0001], $\alpha$, $\beta$, $\gamma$ are tuned within [3, 1, 0.3]. Our $f$ model is a standard dense neural network consisting of eight layers, activated by Tanh functions. The dimensions of each layer are [3, 60, 60, 60, 60, 60, 60, 60, 60, 8]. Our $Q$ model is of a similar architecture of [3, 20, 20, 20, 20, 20, 20, 20, 20, 8]. $\phi(\text{u})$ is determined by the sparse regression of derivatives of all weather factors to space and time up to the second-order, with certain terms that appear in traditional NWP and physical equations fixed as a priori.
>
> 2. __Disagreement to the motivation of PhyDL-NWP.__
>
> Thank you for the comment. Respectfully, we do not agree with the critic that ML can generalize well in weather forecasting tasks. While we believe that philosophical debate is useful, whether our model actually works is most important. Supported with extensive experiments, and additional experiments provided in response to Reviewer HSXq and Reviewer W7oo, we have undoubtedly demonstrated the effectiveness and efficiency of our model. In fact, based on extensive studies of deep learning models in the field, even including large models such as Pangu-Weather, GraphCastNet, etc., one would not be certain that ML can generalize well in real-world situations. For example, Pangu-Weather failed to predict the trajectory of Typhoon Doksuri in 2023, which landed in mainland China and caused incalculable losses. In fact, people find that Pangu's forecast of the landing site is still less accurate than the numerical model, i.e., NWP. If ML models can indeed generalize well, this would not happen, and we could have saved billions of dollars and many lives. Moreover, as an interdisciplinary team focusing on weather prediction research, we have done many simulations but unfortunately found that deep learning models are generally unreliable, especially for long-term prediction. On the other hand, NWP results are relatively better. It is a hard truth, but the status quo of the field. Based on this, we strongly urge the need to incorporate physical insights and combine NWP into deep learning models.
>
> 3. __Complicated "state-of-the-art" architectures are not important.__
>
> Thank you for the comment. Respectfully, based on our experiments, we do find that SOTA architectures are important. Moreover, the development of large models (such as Pangu-Weather, Graphcast, ClimaX, FourCastNet, NowCastNet, etc.) and their amazing results also suggest that model architectures and capacities do matter. For example, AFNO, the backbone of FourCastNet, performs much better than ConvLSTM. Before 2022 when there is no research on large models for weather prediction, there is almost no report that any deep learning model has ever had a comparable result to NWP, even though global weather data is available decades ago. If input and output are the only important things, we should have seen remarkable results long before 2022, maybe even before transformers are invented. Therefore, we respectfully disagree with the reviewers’ critics.
>
> In addition, our point of view regarding the reason PINN won’t work may have been misunderstood. As explained in Sec. 3.3, “In particular, $f_\theta$ is only trained on historical coordinates, while anywhere outside the bounds of where the model was trained is completely unknown to $f_\theta$.” PINN itself also faces the problem of generalization, since it is a deep learning model itself. In contrast, the setting of leveraging historical data to predict future data and the SOTA architectures are much more favorable, which are strong inductive bias and help improve the model accuracy. We hope to combine the advantages of both physics and SOTA deep learning formulation, which motivates our work.

---

> ### Author Response · Authors · 2023-11-22
> **Rebuttal to Reviewer MMEb - Part 2**
>
> 4. __Natural images have more non-local spatio-tepmoral dependencies.__
>
> Thank you for the comment. Respectfully, we partially disagree with your argument. First, weather prediction generally has multiple channels, if not hundreds of channels in theory, while natural images only have 3 dimensions. It might be correct to say that Computer Vision models can process spatio-temporal dependencies as well, with advanced architecture design. In the paper, we only wish to highlight that, weather factors represent spatio-temporally correlated physical process, sometimes explained by chaos theory. For example, the famous Butterfly effect explained that even with very long distance and time span, we may find causation between two distanced coordinates on earth. This degree of non-local dependency, is hardly, at least not usually, the case that can be considered in natural images. To avoid controversy or misunderstanding, however, we are willing to give up that statement in the paper. We thank the reviewer for their input and encourage the reviewer to take a new review of our updated manuscript.

---

### Official Review · Reviewer_HSXq · 2023-11-04

**Soundness:** 2 fair
**Presentation:** 2 fair
**Contribution:** 2 fair
**Rating:** 5
**Confidence:** 4

**Summary:**

The paper attempts to model the meteorological dynamics with physics guided deep leaning method.  Authors propose a physics guided deep learning framework, which can be combined with existing deep learning networks. Experiments are conducted on real world datasets to validate the model.

**Strengths:**

1. First of all, weather forecasting is a very important problem and should get more attention from the ai community to defending the climate change.
2. Generally, I like the idea of combining physics mechanism with deep learning to improve the performance and generalization ability of ai methods. The proposed method seems reasonable.
3. Experiments are conducted on the ERA5 dataset, which is one of the most high-quality weather data. The results show that the proposed method can obtain performance gain for both downscaling and forecasting tasks.

**Weaknesses:**

1. The writing of the paper could be further improved, especially the use of symbols. For example, what is (u, v, w) in figure 1, is Q the same as Q_pi? What is epsilon?
2. One the mentioned advantage of physics guide is physical consistent. However, it is not clear how to measure physical consistent, do you mean the analysis in sec 4.3?
3. There are some important recent works and background information missed in the related work part. Section 2.1 missed some recent progress in weather forecasting such as GraphCast, ClimaX, and FengWu. The relate work part also lacks discussions about PINNs.
4. It is also not clear to me what is the core technique contribution of this paper when aligning it to the broad PINN family.
5. Regarding the experiments, it is hard to compare this work with existing deep learning works since it is conducted in the special region. It would be good to present the results in the full ERA5 dataset or a smaller resolution (e.g., weatherbench) if the resource is limited.
6. The assumption that deep learning model is accurate enough does not hold in this paper.
7. It is not clear about the detailed train, validation, and test settings. There is also no code shared for reproduce ability checking.

**Questions:**

please check the weakness part.

---

> ### Author Response · Authors · 2023-11-22
> **Rebuttal to Reviewer HSXq - Part 1**
>
> We thank the reviewer for their time and constructive comments. We have updated the manuscript to improve organization, results interpretation, and add a new experiment in Section 4.2. We thoroughly address your concerns below. We hope the reviewer’s score can be updated to reflect the significance, novelty, and timeliness of our study. Due to the space limit, we will address the rebuttal in several parts.
>
> 1. __The writing of the paper and especially the description of symbols.__
>
> Thank you for raising this point. We have made substantial revisions. We encourage the reviewer to look at the updated manuscript for further evaluation.
>
> 2. __The measurement of physical consistency.__
>
> Thank you for the comment. In our paper, we consider “physical consistency” as the model’s alignment with existing physical principles. Therefore, we use Sec 4.3 to analyze how well our data-driven discovery of physical mechanism can align with the existing equations. We exclusively encode a priori PDE terms into our PhyDL-NWP framework, which ensures that relevant terms present as a constraint to improve physical consistency, as explained in Sec. 4 and Sec. 4.3. As shown by the empirical results, not only the a priori terms are discovered, but also some new terms are discovered as a supplement to the existing equations.
>
> 3. __Related works of recent progress and core contributions to the broad PINN family.__
>
> Thanks for the comment. We have expanded and reorganized the related work section. Especially, we directly name certain models to be more specific about each baseline’s setting and its difference to our approach. In addition, we improve the presentation to indicate our relationships to PINN family.
>
> Our novel model design is partially based on the physical constraint loss used by PINN, but also expands and improves the PINN to address the problem of weather downscaling and forecasting, collaborating with meteorology experts and physicists. The weather prediction task is highly complicated, with many weather factors and physical processes in the play, easily influenced by local variations like change of boundary conditions, small-scale phenomena like microclimates and external forces like heat from the sun. Many of these vital factors, which have huge impacts in the first-principle equations, are missing in the data due to difficulties to measure and quantify. For example, as Eq. 14 in Page 9 shows, the effect of friction characterizing air viscous resistance cannot be represented by any of the existing data elements. The nuances make the physical equation incomplete. Therefore, it is intractable to rely on a fixed PDE, while PINN directly uses a known PDE to guide optimization under all circumstances, which could be hard to adapt to the real world.
>
> To address this concern, we do not rely on any known physical equations with PINN to guide prediction, but propose a data-driven approach to adaptively consummate the first-principle physical equation to explain the physical mechanism that drives the weather prediction. Our approach has the potential of discovering the intricate interplay between various weather factors that is previously ignored, as an adaption to various conditions in different areas. We not only complete the tasks of weather downscaling and forecasting, but also provide insights of the nuances between different climates at any continuous spacetime. Our model is of great value to the meteorology community, as verified by the physicists we are collaborating with.
>
> Furthermore, we propose a latent force term $Q_\pi$ as a parametrization term in the equation, following the parametrization strategy [1] widely adopted by meteorology experts to supplement the forces that cannot be represented by the selected explicit PDE terms. All of these novel model designs differ from existing works in the PINN family. There is almost no work that successfully applies PINN for weather prediction. We believe that our approach fills this gap in time and provides a feasible way to improve our understanding of the physical mechanism of climate and improve the deep learning models’ performances, which is well-supported by our promising experimental results. We wish to highlight that our PhyDL-NWP framework only contains 55 thousand parameters, as shown in Table 7 in the appendix, which is about 1000 times lighter than some large models and extremely efficient to train. In our experiments, the training of deep learning model usually takes 20~50x more time than obtaining the PDE we need. Our contribution is not trivial and is of great value to not only the meteorology community but also the representation learning community. When tackling similar scientific tasks that also involve complex interplay between variables and insufficient data measurement of the nuances, our work will provide a valuable reference.
>
> [1] Warner, Thomas Tomkins. Numerical weather and climate prediction. Cambridge University Press, 2010.

---

> ### Author Response · Authors · 2023-11-22
> **Rebuttal to Reviewer HSXq - Part 2 (Updated Experiment)**
>
> 4. __Comparison to baselines with ERA5 or WeatherBench dataset.__
>
> Our Ningxia and Huazhong datasets are from ERA5 as explained in Appendix A.1, just like WeatherBench. We select the current three local datasets to highlight our model’s ability to provide supplement to existing physical equations even in a high-resolution local area. Especially, we test our approach on challenging and fine-granular hourly data with 0.25 degrees of spatial resolution and diverse continental climate in arid areas as represented by Ningxia dataset and mild humid oceanic climate as represented by Ningbo dataset. At this scale, weather factors are more likely to be disturbed by local variations like change of boundary conditions and small-scale phenomena like microclimates. WeatherBench, regridded the original 0.25 degree of resolution to the degrees of 5.625, 2.8125, 1.40625, which are much coarser than the data we use. The global weather forecasting based on low-resolution data is of less relevant to our work. We want to highlight our model’s supplement to physical first-principle mechanism and combine such an updated mechanism to guide deep learning models. Therefore, experiments on high-resolution datasets are necessary to test whether our PhyDL-NWP can work even in a relatively noisy environment and can work for long-term medium-range forecasting. We follow a similar setting as in NowCastNet, which also incorporates physical mechanism and focuses on the weather prediction of local regions.
>
> We appreciate the suggestion and include an additional experiment with WeatherBench. To make fair comparison, we use the 5.625 degree of spatial resolution and 6 hours setting to preprocess the data. We use 10 years of training data from 2006 to 2015, validation data in 2016 and test data in 2017-2018. We select ground temperature (t2m), atmospheric temperature (t), geopotential (z) and ground wind vector (u10, v10) as the weather factors. We apply the normalization to these weather factors and used their recommended training regime and hyperparameters. We use this data to perform weather forecasting and compare with baselines.
>
> | Variable | Time Steps (hours) | AFNO | AFNO+ | AFNO | AFNO+ |
> |----------|-------------------|-------------|---------|-------------|---------|
> |          |                   | RMSE        | RMSE    | ACC         | ACC     |
> | t2m      | 6                 | 1.25        | 1.18    | 0.95        | 0.98    |
> |          | 12                | 1.49        | 1.32    | 0.93        | 0.97    |
> |          | 18                | 1.64        | 1.43    | 0.91        | 0.96    |
> |          | 24                | 1.80        | 1.62    | 0.89        | 0.96    |
> |          | 48                | 2.45        | 2.14    | 0.82        | 0.92    |
> | t        | 6                 | 1.20        | 1.18    | 0.97        | 0.97    |
> |          | 12                | 1.50        | 1.36    | 0.95        | 0.97    |
> |          | 18                | 1.75        | 1.47    | 0.92        | 0.96    |
> |          | 24                | 1.99        | 1.68    | 0.88        | 0.96    |
> |          | 48                | 2.78        | 2.30    | 0.84        | 0.93    |
> | z        | 6                 | 142.5       | 138.3   | 0.98         | 0.99    |
> |          | 12                | 201.8       | 167.0   | 0.97        | 0.99    |
> |          | 18                | 256.0       | 205.2   | 0.96        | 0.99    |
> |          | 24                | 309.0       | 250.6   | 0.94        | 0.98    |
> |          | 48                | 477.2       | 424.1   | 0.86        | 0.94    |
> | u10      | 6                | 1.50        | 1.42    | 0.94        | 0.96    |
> |          | 12                | 1.88        | 1.76    | 0.92        | 0.94    |
> |          | 18                | 2.12        | 1.95    | 0.90        | 0.93    |
> |          | 24                | 2.34        | 2.15    | 0.87        | 0.91    |
> |          | 48                | 2.98        | 2.63    | 0.72        | 0.84    |
> | v10      | 6                 | 1.56        | 1.55    | 0.92        | 0.92    |
> |          | 12                | 1.81        | 1.64    | 0.90       | 0.91    |
> |          | 18                | 2.22        | 1.89    | 0.87        | 0.90    |
> |          | 24                | 2.57        | 2.13    | 0.84        | 0.88    |
> |          | 48                | 3.46        | 2.90    | 0.69        | 0.75   |
>
> It shows our model’s performance is more significant on coarser-grained weather data. Since we use coarse resolution of WeatherBench data with prediction in 48 hours, it is normal that the ACC metrics are higher than the ones reported with our original two datasets. This might answer the reviewer’s other question regarding the accuracy of deep learning models in our paper in general. In contrast, our paper considers more challenging datasets with higher resolution and long-term prediction setting.

---

> ### Author Response · Authors · 2023-11-22
> **Rebuttal to Reviewer HSXq - Part 3**
>
> 5. __Deep learning model is not accurate enough in this paper.__
>
> Thank you for the comment. As Tables 2 and 3 have shown, deep learning models (Bi-LSTM-T, Hybrid-CBA, ConvLSTM, AFNO, MTGNN, and MegaCRN) clearly have lower RMSE than NWP. It is true that on the metric of ACC, NWP is the best when we consider the 7-day weather forecasting. However, note that, as visualized in Figure 3 in the main paper and Figures 5-8 in the Appendix, when we consider the short-term weather forecasting, especially within 48 hours, the AFNO’s ACC curve is clearly above 0.526, showing superior performance against non-deep learning NWP. NWP is based on numerical simulation, therefore has a fixed value at any certain time regardless of the prediction time and the length of historical values used. Furthermore, as we explained in response to Q4, when we use coarse resolution of WeatherBench data, the ACC metrics are much higher than the ones reported with our original two datasets. This is because a coarse resolution dataset is relatively not likely to be severely affected by local variations and noises. In conclusion, deep learning models in our paper are in fact quite accurate. The seemingly lower performance is only due to the challenging data we use and the long-term prediction setting we apply.
>
> 6. __The setting of train, validation, and test. The code.__
>
> Thanks for the comment. Due to the limited space to present our paper, we explained in Sec. 4.2 on Page 7 that “More details of the datasets are given in Appendix A.1”. We are glad to provide the core code in an anonymous link. https://anonymous.4open.science/r/123-C702/README.md

---

### Author Response · Authors · 2023-11-23
**Response to all reviewers**

Dear reviewers,

We would like to thank you all again for your feedback. We are grateful that most reviewers agree our work is original, well-supported by extensive experiments, targeting on an important issue. As the interactive rebuttal period is short, we sincerely hope you can look through them and participate in the discussion to help us make this paper better. Since most of the concerns are surrounding the experiments, we have added many new experiments as suggested and made substantial revision. Meanwhile, we believe we have addressed all the concerns, and we are willing to answer any further questions if there are still concerns or other concerns.

Thanks,
The authors of the paper

---

### Meta-Review · Area_Chair_n4CP · 2023-12-08

**Metareview:**

The paper presents a method that combines deep learning with traditional numerical methods for weather modeling. Various components are modeled via neural networks representation of partial differential equation (PDE). The experiments focus in whether the proposed approach does well on weather downscaling and betters existing weather forecasting models.

One of the reviewers raise the issue of "Small-scale datasets targeting on regions of China" - I agree with the reviewers that the choice of region should not have any bearing on acceptance, provided that the paper meets the acceptance threshold both for presentation and experimentation.

The biggest criticism of the work is indeed presentation and most of the reviewers raise this point. The spectrum of criticism ranges from unclear motivation, hard to understand description of the method, contribution, to lack of clarity on significance of experimentation. And it is for this that I recommend rejection.

The paper can be improved significantly in terms of clarity and presentation, and I suggest authors to follow the reviewer's comment and resubmit to an alternate venue.

**Justification For Why Not Higher Score:**

Most of the reviewers are consistent in the criticism and lack of any reviewer strongly arguing for acceptance of the paper.

**Justification For Why Not Lower Score:**

N/A

---

### Decision · Program_Chairs · 2024-01-16

Reject